# Spatially resolved proteomic map shows that extracellular matrix regulates epidermal growth

Jun Li [1,5✉], Jie Ma[2,5], Qiyu Zhang[1,3,5], Huizi Gong[1,3], Dunqin Gao[3], Yujie Wang[3], Biyou Li[2,4], Xiao Li[2], Heyi Zheng[1], Zhihong Wu[3], Yunping Zhu [2,4✉] & Ling Leng [3✉]

Human skin comprises stratified squamous epithelium and dermis with various stromal cells and the extracellular matrix (ECM). The basement membrane (BM), a thin layer at the top of the dermis, serves as a unique niche for determining the fate of epidermal stem cells (EpSCs) by transmitting physical and biochemical signals to establish epidermal cell polarity and maintain the hierarchical structure and function of skin tissue. However, how stem cell niches maintain tissue homeostasis and control wound healing by regulating the behavior of EpSCs is still not completely understood. In this study, a hierarchical skin proteome map is constructed using spatial quantitative proteomics combined with decellularization, laser capture microdissection, and mass spectrometry. The specific functions of different structures of normal native skin tissues or tissues with a dermatologic disease are analyzed in situ. Transforming growth factor-beta (TGFβ)-induced protein ig-h3 (TGFBI), an ECM glycoprotein, in the BM is identified that could enhance the growth and function of EpSCs and promote wound healing. Our results provide insights into the way in which ECM proteins facilitate the growth and function of EpSCs as part of an important niche. The results may benefit the clinical treatment of skin ulcers or diseases with refractory lesions that involve epidermal cell dysfunction and re-epithelialization block in the future.

[1] Department of Dermatology and Venereology, Peking Union Medical College Hospital, Chinese Academy of Medical Sciences and Peking Union Medical College, Beijing, China. [2] State Key Laboratory of Proteomics, Beijing Proteome Research Center, National Center for Protein Sciences (Beijing), Beijing Institute of Lifeomics, Beijing, China. [3] Stem Cell and Regenerative Medicine Lab, State Key Laboratory of Complex Severe and Rare Diseases, Translational Medicine Center, Peking Union Medical College Hospital, Chinese Academy of Medical Sciences and Peking Union Medical College, Beijing, China. [4] Basic Medical School, Anhui Medical University, Anhui, China. [5] These authors contributed equally: Jun Li, Jie Ma, Qiyu Zhang. ✉email: lijun35@hotmail.com; zhuyunping@ncpsb.org.cn; lengling@pumch.cn

The skin, as the first and largest barrier organ in the body, has a tough and pliable structure that enables it to adapt to external conditions by quickly repairing mechanical, chemical, and biological injuries. In mammals, the skin comprises several distinct cell populations that are organized into the following essential layers: the epidermal, dermal, and hypodermal (subcutaneous fat) layers. The epidermis is the outermost layer that is composed of stratified cell layers, which are maintained by keratinocytes, including stem cells and an abundance of mature cells[1]. The basal layer (BL) of the epidermis is the location of undifferentiated proliferative progenitor cells, that is, epidermal stem cells (EpSCs), which express the keratins K5 (KRT5) and K14 (KRT14)[2]. These progenitors replenish BL via self-renewal, progressively migrate upward, and differentiate into mature keratinocytes, which express KRT1, KRT10, and involucrin and are located in the granulosum and spinosum layers. Finally, the outer layer stratum corneum (SC), which consists of terminally differentiated and dead SC cells, serves as a scaffold for the lipid bilayers that comprise the epidermal barrier on the skin surface[3,4]. The commitment of EpSCs to a specific lineage is regulated by a combination of intrinsic and extrinsic mechanisms. However, there is a lack of detailed information about the molecular compositions and regulatory functions of specialized proteins that are localized in different zones during the dynamic processes of skin development, homeostasis, and regeneration for wound healing[5,6].

The extracellular matrix (ECM) is a complex three-dimensional microenvironment that provides cells with many chemical and biophysical signals that are necessary for cellular function. The ECM microenvironment provides skin cells with a resilient, viscoelastic, cushioned environment that is conducive to cell adhesion, proliferation, and migration[7]. Multiple proteins play important roles in the ECM. The dermis usually contains an abundance of ECM proteins, which behave as an in situ scaffold that contains multiple components, including collagens, glycosaminoglycans, hyaluronic acid, fibronectin, elastin (ELN), laminin, and other ECM proteins[8]. In addition, there is a thin layer of specialized ECM proteins at the junction of the dermis and epidermis. These proteins include type IV and VII collagens, laminin, perlecan, growth factors, and other ECM proteins[9] as part of the basement membrane (BM). The BM links the keratin intermediate filaments of basal keratinocytes with collagen fibers in the superficial dermis (SD), influences biological interactions between EpSCs and the ECM, coordinates the actin and microtubule network, and establishes cell polarity. Therefore, basal EpSCs can correctly proliferate and differentiate to produce the appropriate tissue structure by forming various cell junctions[10,11].

After an injury occurs, EpSCs replicate and differentiate into the mature epidermis and regenerate hair follicles. Wnt/β-catenin is involved in maintaining EpSCs by regulating EpSC proliferation during both skin homeostasis and wound healing[12–14]. Several phosphorylation sites in β-catenin can mediate β-catenin degradation, such as β-catenin on S33/S37T41[15,16]. Canonical WNT proteins inhibit the expression of GSK3β kinase, resulting in the accumulation of stable, unphosphorylated β-catenin proteins[17]. β-catenin acts as a nuclear cofactor for the LEF1/TCF family of DNA-binding proteins to activate canonical WNT downstream pathways[17,18]. The BM is responsible for wound healing, revascularization, cell development, morphology, and immunity during wound repair by regulating EpSCs via integrin receptors or by regulating the interactions between growth factors and their receptors[19]. However, the mechanism by which the ECM of the BM, as a niche for stem cells, maintains skin homeostasis and promotes wound repair by regulating EpSCs remains poorly understood.

The spatial expression of proteins is critical for identifying the exact localization and function of proteins in tissues. In this study,

using a combination of decellularization, laser capture microdissection (LCM), and mass spectrometry (MS), proteins from six skin layers - SC, granular-spinous (GS), BL, BM, SD, and deep dermis (DD) - were isolated and enriched. As a result, a stratified developmental lineage proteome map of human skin that incorporates spatial protein distribution was constructed. In the BM, TGFβ-induced protein ig-h3 (TGFBI), an ECM glycoprotein, can enhance the proliferation of EpSCs and promote the process of wound healing. Most interestingly, TGFBI shows the potential to treat patients with refractory wounds and may benefit clinical practice.

## Results

**Spatial proteome architecture of stratified human skin.** Epidermal development proceeds through the BL, which contains proliferative and undifferentiated epidermal cells (KRT14+COL17A1+PLEC+), to the mature epidermal layer (KRT10+) (Fig. 1a). Samples from six skin layers (SC, GS, BL, BM, SD, and DD) were obtained using LCM combined with a decellularization strategy that was established by our group[20] (Fig. 1b, Supplementary Fig. 1, and Supplementary Data 1) to conduct an in-depth study of skin function according to the stratified structure. Using spatial MS, we analyzed the protein abundance in each layer and combined these data with precise anatomical localization to construct a spatial proteome map of human skin (Fig. 1c). As a result, 4896 proteins were identified, including 4686 in SC, 4676 in GS, 4556 in BL, 3775 in BM, 4677 in SD, and 4443 in DD (Fig. 1d, Supplementary Fig. 2a, and Supplementary Data 2). The reference proteome was highly dynamic, spanning approximately eight orders of magnitude, as determined by measuring protein abundance (Supplementary Fig. 2b). When correlating the proteins in the layers, we found that the proteomes of the six skin layers could be grouped into the following skin sections: the epidermis section that included the SC, GS, and BL, the BM section, and the non-BM dermis section that was composed of the SD and DD (Fig. 1e and Supplementary Fig. 2c).

Cellular component analysis showed significant differences between the epidermis, BM, and dermis sections (Fig. 1f). The biomechanical components of keratinocytes, such as intermediate filament and keratin filament components, were mainly enriched in the epidermis. Keratins, which make up the cytoskeleton of epithelial cells, were highly expressed in the epidermis, especially in the SC (Fig. 1g). Further analysis showed that the keratins related to the epidermal barrier and keratinocyte differentiation, such as KRT78, KRT79, KRT1, KRT10, and KRT80, were highly expressed in the SC (Supplementary Fig. 3a). The EpSC marker keratins, such as KRT14, KRT15, and KRT5, were primarily expressed in BL. On a larger scale, keratins involved in hair follicles, including the inner hair root sheath (IRS) and outer hair root sheath (ORS), were mainly enriched in the epidermis. However, keratins and keratin-associated matrix related to hair and nails were mainly enriched in the dermis (Supplementary Fig. 3a). These results indicate that the specific expression of keratins, the main cytoskeletal proteins in epithelial cells, is related to the cell type, stage of development, and spatial localization. The components of the cell–cell junctions (desmosomes) and cell–matrix junctions (hemidesmosomes) of keratinocytes were also mainly enriched in the epidermis (Fig. 1f). On the other hand, ECM components, such as collagen trimers and elastic fibers that control the mechanical and elastic properties of the skin, were mainly enriched in the dermis, including the BM section (Fig. 1f). The expression levels of core ECM proteins, including collagens, glycoproteins, and proteoglycans, increased gradually from the epidermis to the DD (Fig. 1g). There was no significant difference in the levels of ECM-associated proteins,

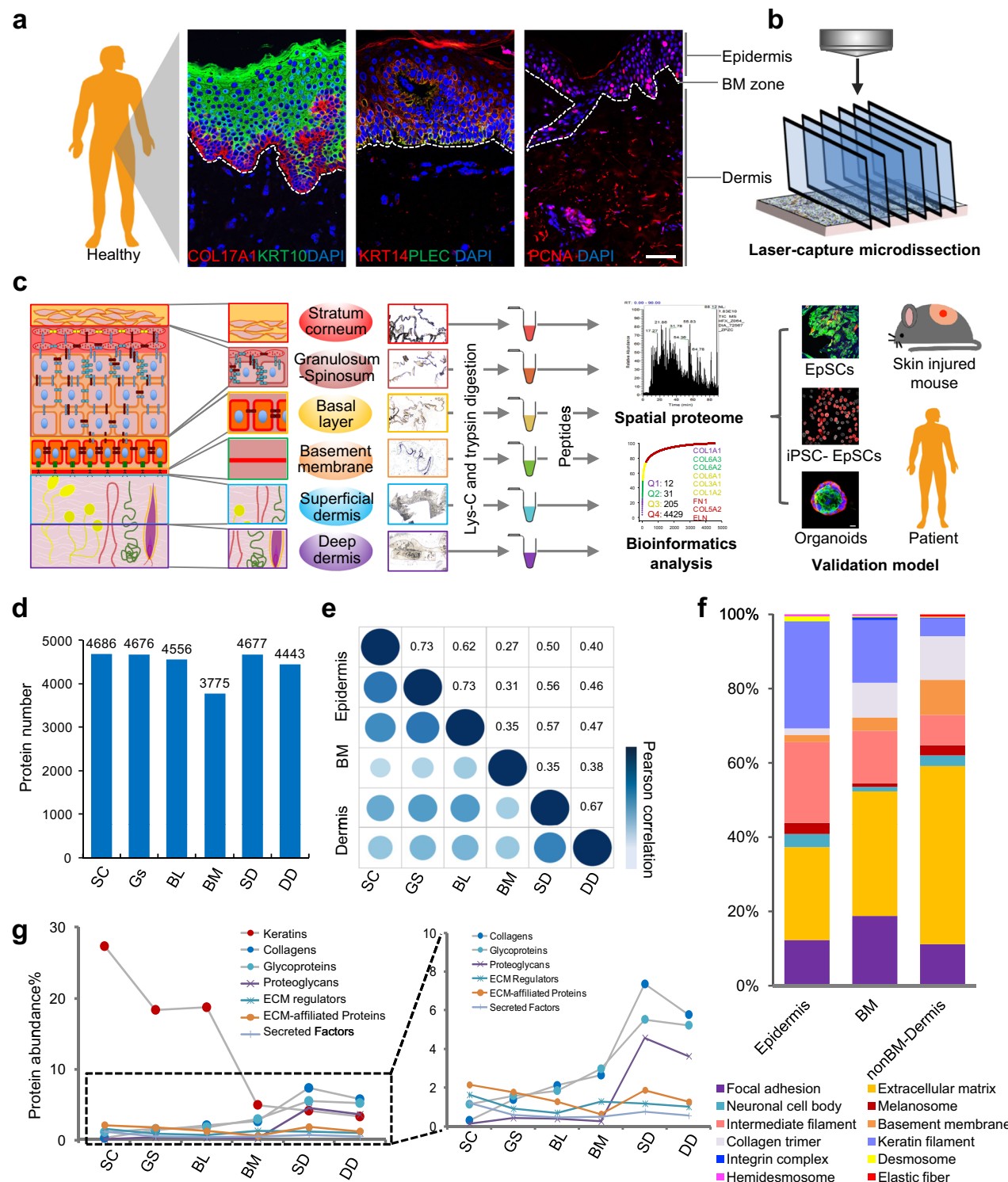

including ECM regulators, ECM-affiliated proteins, and secreted factors, in different skin layers. These results indicate that the supportive role of core ECM proteins in the dermis and those different regulatory proteins and other factors may play essential roles in each skin layer.

A hierarchical proteomics identification system was successfully established and used to construct the proteome map of human skin. The particular components of the six stratified cell layers were identified. These results enabled us to study the specific functions of the six layers of human skin tissues.

**Spatial functional characteristics of stratified human skin.** To identify the boundaries that demarcate the key distinct features and their biological significance, hierarchical clustering analysis (HCA) and Gene Ontology (GO) analysis were performed, and six protein modules characterizing strata of the skin with distinct functions were identified (Fig. 2a). These region-specific proteins were linked to functional differences in the skin (Fig. 1a). In this study, the epidermis was divided into the following parts according to the degree of epidermal cell development: SC, GS, and BL. Module 1 represents the highest concentration of

**Fig. 1 Quantitative proteome profiling of spatially distinct protein signatures in normal human skin. a** Immunofluorescence of mature epidermal markers (KRT10, green), basal stem cell markers (COL17A1 and KRT14, red), and proliferation markers (PCNA, red) (scale bar: 50 μm). The dashed line shows the basement membrane (BM) zone of the skin tissues. The experiment was repeated three times. **b** Laser capture microdissection was used to dissect and process the cells. **c** Schematic of the experimental workflow of the quantitative proteomics, bioinformatics, and biological validation analysis of six layers of human skin samples based on the skin structure, including stratum corneum (SC, red), granulosum-spinosum (GS, orange), basal layer (BL, yellow), basement membrane (BM, green), superficial dermis (SD, blue), and deep dermis (DD, purple). The dissected samples from each structure were pooled, digested, and analyzed using LC–MS/MS. The proteome results were verified using various cell and animal models, including primary epidermal stem cells (EpSCs), induced pluripotent stem cell (iPSC)-derived EpSCs, an epidermal organoid model, a mouse injury model, and patient skin tissue samples. **d** Number of proteins identified in SC, GS, BL, BM, SD, and DD. **e** Region-dependent analysis of the skin proteomes based on the correlation matrix between the following layers: SC, GS, BL, BM, SD, and DD. **f** Cellular component analysis of proteins from three parts of skin, including the following layers: epidermis (SC, GS, and BL), BM, and dermis (SD and DD). **g** Protein abundance percentages of keratins, collagens, glycoproteins, proteoglycans, extracellular matrix (ECM) regulators, ECM-affiliated proteins, and secreted proteins across six skin layers. The percentage was calculated by dividing the summed intensities of the proteins of interest by the summed intensities of all proteins identified in a specific skin layer. Source data are provided as a Source Data file.

---

proteins in the SC among the six layers (Fig. 2a), which were involved in various aspects of aging, such as cornification, keratinization, lipid metabolism, proteolysis, and programmed necrotic cell death, and these proteins were also involved in the process of resistance to external stimulation, such as regulation of water loss, production of antibacterial peptides, and response to light intensity. It is possible that the biological functions of SC include the termination of all metabolic activity and the loss of cytoplasmic organelles to produce the epidermal barrier on the skin surface[21]. Modules 2 and 3 represent the proteins that are highly expressed in mature (GS) and proliferative (BL) epidermis, respectively (Fig. 2a). The proteins enriched in the GS, which are located close to the SC, were capable of functions similar to the SC, such as aging (mitochondrial fragmentation and peroxisome organization) and defense against bacteria, and these proteins were explicitly involved in processes of nervous system development, such as neuronal differentiation, exocytosis of neurotransmitters, and Schwann cell migration. However, the proteins highly expressed in basal epidermal cells performed more specific functions, such as involvement in prostaglandin production, immune system processes, exocytosis, neurofilament cytoskeleton organization, and wound healing. The proteins enriched in the BL were involved in the biological processes of epidermal cell differentiation (e.g., hair follicle maturation) and proliferation (e.g., hair cycle phase), indicating that the BL is mainly responsible for epidermal cell development.

There is a lineage gradient in the development of epidermal cells from the BL to the SC. In our study, proteins that exhibit different trends in expression abundances in epidermal cells from BL to GS and SC were analyzed. Proteins with increased abundances were enriched in the biological processes of keratinocyte differentiation, lipid metabolism, histone phosphorylation, DNA repair, and PPAR signaling pathway, while those with decreased abundance from the BL to the GS and SC were mainly involved in ECM organization, cell-matrix adhesion, angiogenesis, the establishment of mitotic spindle orientation, and the NF-κB signaling pathway (Supplementary Fig. 3b). The markers of the mature epidermis (KRT1, KRT10, and IVL) were highly expressed in SC and GS. In contrast, the EpSC markers (KRT14, KRT5, COL17A1, and P63) and a proliferation marker (PCNA) were highly expressed in the BL (Fig. 2b, c). The proteins associated with cell–cell junctions, including desmosome complexes (DSC1, DSG1, DSC3, PKP1, DSP, DSG3, DSC2, and DSG2) and E-cadherin (ECAD), were identified in the mature epidermal cells of SC and GS. Cell-matrix interactions, including hemidesmosome complexes (COL17A1, ITGA6, ITGB4, and PLEC), focal adhesion complexes (ITGB1 and ITGA3), and important receptors (CD44 and EGFR), were expressed in the BL (Fig. 2b, c). Among the top 10 proteins that were highly expressed

in the epidermis, DNA repair-associated proteins (H4C1 and H2AC groups) and COL7A1 secretion required protein (MIA3) were the dominant proteins in the epidermis, in addition to keratins (Supplementary Fig. 2b), indicating that the epidermis is vital for DNA repair.

The proteome of the dermis at different depths was analyzed. The results showed that the proteins enriched in the dermis were involved in endocytosis and exocytosis, immune responses, and coagulation (Fig. 2a). In addition, the highly enriched proteins in DD were involved in the regulation of neuronal synaptic plasticity and neurotransmitter secretion. Neuronal development-associated proteins (KIF20B and HMCN1) were among the 10 most highly expressed proteins in the SD (Supplementary Fig. 2b). The classical components of the dermis were also identified. For example, the ECM of type I, III, V, and VI collagens, fibronectin 1 (FN1), and ELN were expressed in the SD and DD (Fig. 2b). ECM proteins, which are located extracellularly, play an essential role in regulating dermal elasticity, water retention, aging, and wound healing. The ECM components of the dermis layers were further analyzed using the decellularization method combined with the Matrisome database. A map of the ECM of the stratified human dermis was constructed, including the ECM proteins identified in BM (277), decellularized SD (280), and decellularized DD (284) (Supplementary Fig. 4a and Supplementary Data 3). There were 261 ECM proteins coexpressed in all these decellularized skin layers; however, the abundances of six categories of ECM proteins between BM and decellularized SD/DD were considerably different (Supplementary Fig. 4b–d).

The BM, which is connected with the BL, is the most important niche for EpSCs of the BL. As EpSCs in the BL determine the fate of epidermal cells (proliferation or differentiation), the ECM components of the BM were investigated. We observed a distance between the ECM proteins in the BM and the decellularized dermis (SD and DD) (Supplementary Fig. 4e). The total intensity of glycoproteins and ECM regulators in the BM was significantly higher than that in the decellularized dermis, indicating their important roles as EpSC niches. Biological analysis showed that the enriched proteins of the BM were mainly involved in defense responses, including the establishment of the endothelial barrier and response to bacteria, maintenance of cell polarity, including actin filament bundle assembly and cell substrate adhesion, as well as the development of the substantia nigra (Fig. 2a). Several classical ECM proteins were identified in BM, such as type IV, VII, and XVII collagens (COL4A2, COL7A1, and COL17A1), laminins (LAMC1 and LAMA3), nidogens (NID1 and NID2), and perlecan (HSPG2) (Fig. 2b, c). Using these results, a spatial proteome map of skin tissues was constructed based on the functional and structural characteristics of the skin. These data were then used to investigate dermatologic diseases.

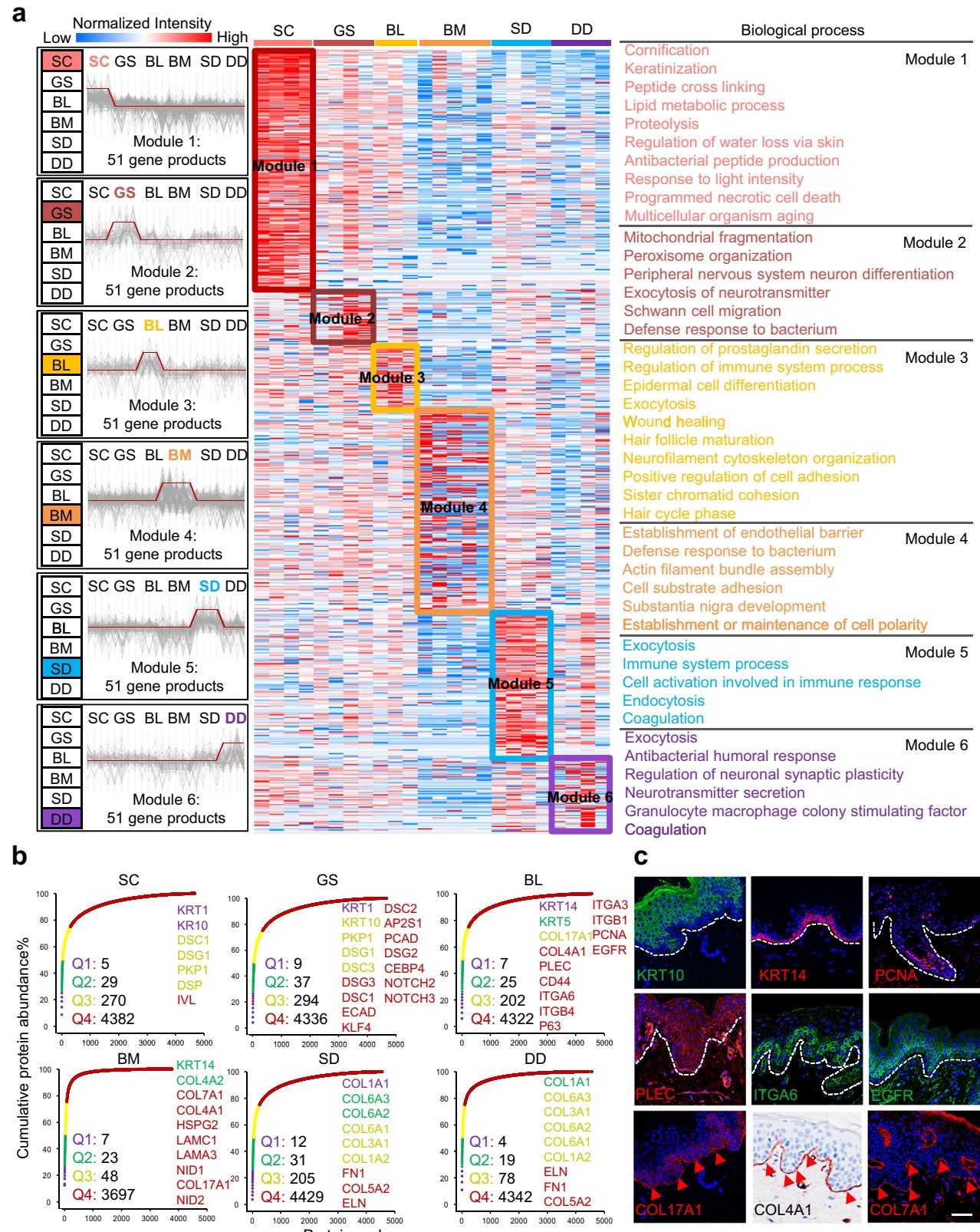

**Epidermal dysfunction in patients with secondary syphilis.** Syphilis, a disease caused by *Treponema pallidum* (*T. pallidum*), is initially characterized by skin lesions and can eventually damage the brain, nerves, eyes, or heart. In this study, *T. pallidum* was mainly detected in the lower epidermis and upper dermis (BL) of skin tissues from patients with secondary syphilis (SSP) (Fig. 3a).

This finding was consistent with that of previous studies, in which the immunohistochemistry-based detection of *T. pallidum* was performed in BL[22–24], which contains EpSCs. It was found that the expression of EpSC markers (KRT14), desmosomes (DSG2), and hemidesmosomes (ITGA6) in the BL, as well as mature epidermal cell markers (KRT10 and KRT1) and desmosomes

**Fig. 2 Spatial functional characteristics of stratified human skin. a** Six protein modules reveal proteome specificity based on skin structure using coexpression analysis. Left panel: The coexpression patterns of the proteins in the six modules. The clusters of proteins associated with similar biological processes are grouped according to the degree of enrichment, as indicated on the right side. Red, orange, yellow, green, blue, and purple bars correspond to the proteins enriched in the six modules. Each column within the skin regions corresponds to a different human skin specimen. The red and blue boxes indicate proteins with increased and decreased abundances, respectively. **b** Cumulative protein abundances for each layer and the total number of proteins constituting the quantiles (Q1-Q4). **c** Immunofluorescence and immunohistochemistry analysis of KRT10, KRT14, PCNA, PLEC, ITGA5, EGFR, COL17A1, COL4A1, and COL7A1 (scale bar: 50 μm). The dashed lines represent the BM zone of the skin tissues. The red arrows refer to the ECM proteins located in the BM zone. The experiment was repeated three times. Source data are provided as a Source Data file.

(DSG1) in the GS, was downregulated in the SSP group compared with the control group, which could account for ulceration and bullous formation in some *T. pallidum* infection cases[25]. To determine the effect of *T. pallidum* on the functions of different epidermal regions in the SSP, proteomic analysis was performed on the skin tissues from the GS (SSP-GS), BL (SSP-BL), and SD (SSP-SD) groups (Fig. 3b). In both the SSP and control samples, a total of 4258, 4268, and 4120 proteins were identified in the GS, BL, and SD, respectively, with significant differences by principal coordinate analysis (PCoA) (Supplementary Data 4 and Supplementary Fig. 5a–d). HCA and GO term analyses were performed on the proteins that were differentially expressed in the BL between the SSP and control samples (Fig. 3c). Immune-associated processes, including neutrophil aggregation, innate immune responses, chemokine production, leukocyte migration, and autophagy, were upregulated in the SSP group compared with the control group, suggesting the presence of an inflammatory response by EpSCs to immune cells in the skin tissue of the SSP. Proteins associated with the cell cycle and morphogenesis processes, ECM organization, cell–cell adhesion, and hemidesmosome assembly were downregulated in the SSP group compared with the control group (Fig. 3c), indicating that epidermal cell dysfunction occurred in the BL. In addition, the canonical Wnt signaling pathway was downregulated in SSP, indicating that the dysfunction of epidermal cells in BL may have been mediated by the Wnt signaling pathway, which is known to be involved in EpSC development during skin homeostasis and wound healing.

To further explore the dysfunction of EpSC development, these proteins that exhibit variable expression patterns from BL to GS in SSP and normal skin were investigated. There were 109 and 99 proteins in which the abundances gradually decreased and increased in the normal group, respectively, while the abundances of these proteins showed opposite trends in the SSP group (Fig. 3d and Supplementary Fig. 5e, f). Biological process analysis revealed that functions associated with epidermal growth, cell migration, immune responses, and apoptosis in the BL, as well as lipid metabolism, vesicle transport, and endocytosis in the GS, were mainly damaged in the skin tissues of the SSP (Fig. 3d). There was also unbalanced expression of keratins in the SSP group (Supplementary Fig. 5f). The keratins associated with the epidermal barrier (e.g., KRT78 and KRT79) and keratinocyte differentiation (e.g., KRT1, KRT10, and KRT3) in the epidermis of the SSP were downregulated compared with those in the control group (Supplementary Fig. 5g, h). Keratins of epidermal stem markers (e.g., KRT14, KRT15, and KRT5) were downregulated in BL and GS (Supplementary Fig. 5i). However, most of the keratins associated with sweat glands, hair follicles, hair and nails, and hair matrix were upregulated in the epidermis (Supplementary Fig. 5i–k). Most of the keratins associated with IRS were downregulated, while those associated with ORS were upregulated in the SSP group compared with the control group (Supplementary Fig. 5l). These results indicate that *T. pallidum* infection could destroy epidermal homeostasis and the expression of several important cytoskeletal components that are essential

for the functions of the epidermis and its appendages. It has been reported previously that BM, as an EpSC niche, plays a critical role in determining the fate of EpSCs[26]. Therefore, we next investigated the role of the ECM in EpSC function in skin tissues.

**TGFBI of BM enhanced EpSC proliferation.** To investigate the ECM proteins in the BM that are important for the regulation of EpSC function, the BM components and hemidesmosome complexes that stayed in contact with the BM and BL were analyzed. Most of the ECM proteins located on the BM and hemidesmosomes were downregulated in the SSP skin tissues (Fig. 3e). Among these proteins, the expression of transforming growth factor-β-induced (TGFBI) was downregulated in both the BL and SD of SSP (Fig. 3e). Consistent with the proteome data, the expression of TGFBI was downregulated substantially around the BM zone (Fig. 3f). An interaction network was constructed to represent the interactions between TGFBI and the proteins identified in the BL (Supplementary Fig. 4f). TGFBI was found to interact with ITGB1 and ITGA3, which are associated with the classical Wnt pathway. Immunofluorescence analysis showed that TGFBI was expressed around the BM region of the fetal and adult skin of humans and the skin of mice (Fig. 4a and Supplementary Fig. 6a). Most of the TGFBI was expressed in the dermis rather than in the epidermis (Supplementary Fig. 6b). TGFBI expression could be induced by treatment with TGFβ1 and was secreted in the extracellular space between fibroblasts (Supplementary Fig. 6c–e).

Next, TGFBI recombinant protein was added to a culture of human EpSCs. Transcriptome analysis showed that the proteins were mainly enriched in the biological signals of the cell cycle and cell proliferation (Fig. 4b, c and Supplementary Data 5). The expression of proliferation markers (PCNA and Ki67) and EpSC markers (P63 and KRT14) increased in the samples with the TGFBI treatment compared with control samples (Fig. 4d, e and Supplementary Fig. 6f, g). However, the expression of proteins in cell differentiation-related pathways (e.g., Notch signaling, NOTCH1 and LAMA5) and the maturation marker of EpSCs (KRT10) were decreased following TGFBI treatment (Fig. 4c, f). In addition, the TGFBI treatment produced a more explicit characterization of the epidermis, with the abundance of mesenchymal-epithelial transition-associated proteins (e.g., CLDN12, OCLN, TJP2, and TJP3) increased and the epithelial-mesenchymal transition-associated proteins (e.g., FN1, LAMA3, VIM, and COL1A1) were decreased compared with that of the control group (Fig. 4g). To further investigate the effect of TGFBI on EpSC proliferation and stemness, induced pluripotent stem cell (hiPSC)-derived EpSCs were used for 30 days of differentiation (Fig. 4h). The expression of the transcription factor P63, stemness marker KRT14, and epithelium marker ECAD in EpSCs was upregulated after the TGFBI treatment compared with that in the control group (Fig. 4h).

According to the interaction network analysis, TGFBI was associated with several Wnt pathway-related proteins (ITGA3, ITGB1, SMAD2, and APP, Supplementary Fig. 4f). XAV939, a potent tankyrase inhibitor, can promote β-catenin degradation

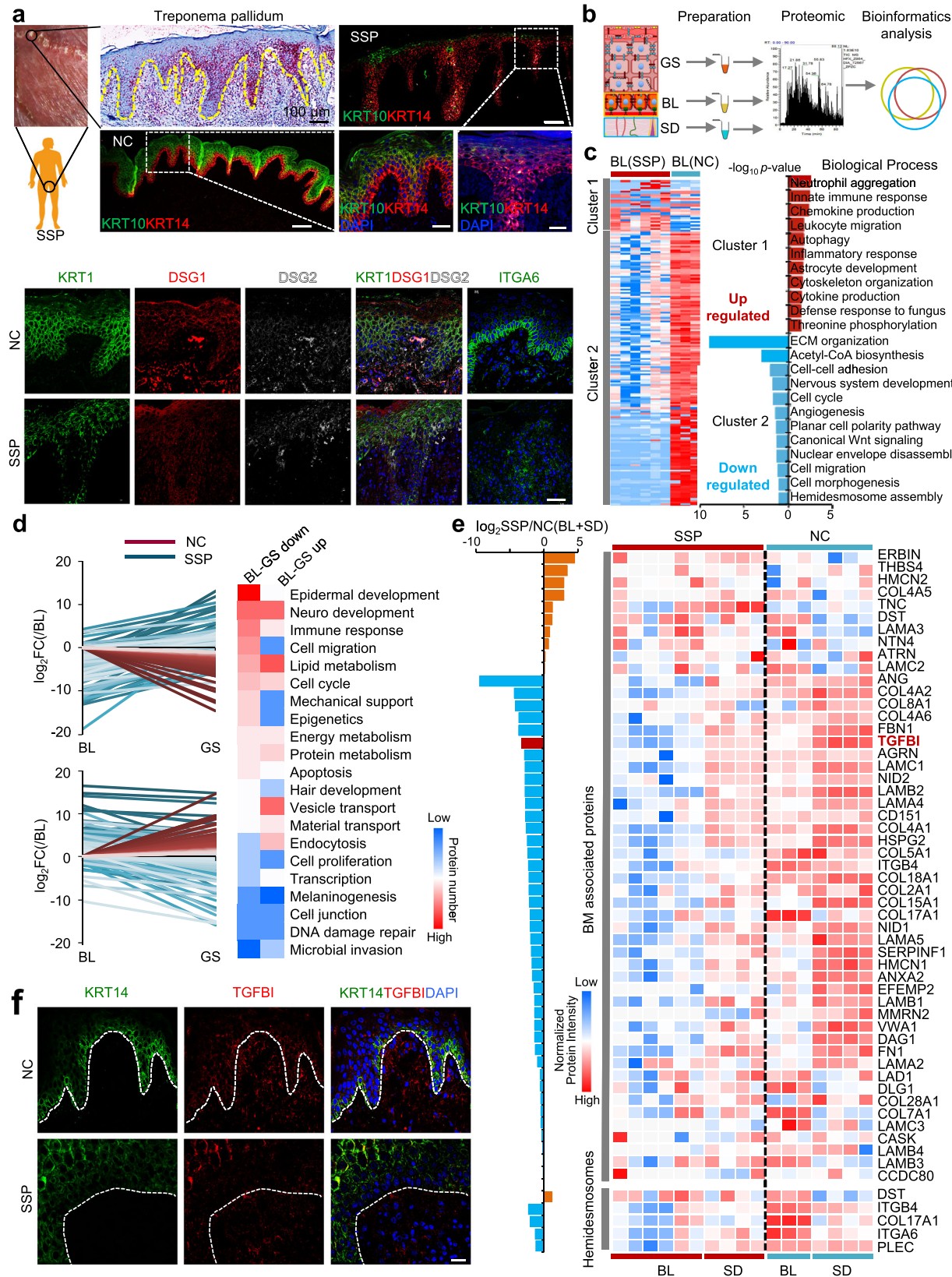

via stabilization of axin by inhibiting tankyrase 1 and tankyrase 2[27], which can help prevent β-catenin from entering the nucleus. After the Wnt inhibitor XAV939 was added, the expression of Ki67, P63, KRT14, PCNA, and ECAD and the number of EpSC clones decreased markedly following the XAV939 treatment, as expected (Figs. 4i–j, 5a–c, Supplementary Figs. 6g, 7a).

Interestingly, it was found that adding TGFBI to the EpSCs with the XAV939 treatment failed to promote the entrance of β-catenin into the nucleus, proliferation ability or stemness of EpSCs, indicating that regulation between the TGFBI and wnt signals occurred (Supplementary Fig. 7b). Furthermore, the expression of the maturation marker KRT10 was upregulated in

**Fig. 3 Epidermal dysfunction of secondary syphilis patients based on spatial proteome profiles. a** Injured foreskin and immunohistochemistry of *Treponema pallidum* from secondary syphilis patients (SSP) (scale bars: 100 and 10 μm). The experiment was repeated three times. Immunofluorescence of KRT14, KRT10, KRT1, DSG1, DSG2, and ITGA6 was analyzed in foreskin tissue of SSP (10 μm). **b** Experimental workflow of quantitative proteome and bioinformatics analysis of the following layers of SSP skin: GS (orange), BL (yellow), and SD (blue). **c** Heatmap of the differentially expressed proteins (DEPs) of BL in the control (NC) and SSP groups. The red and blue boxes indicate proteins with increased and decreased abundance, respectively. The biological processes of proteins in Clusters 1 and 2 are grouped according to the -$\log_{10}$ p value of the degree of enrichment. The DEPs between SSP ($n = 6$) and control ($n = 3$) samples were determined based on the p value of a moderated t test using the R package Limma: p value < 0.01 and $\log_2$ SSP/Control >1 (Cluster 1: upregulated), and p value < 0.01 and $\log_2$ SSP/Control <−1 (Cluster 2: downregulated). **d** Expression levels of protein abundance decreased and increased from the BL to the GS in the control group (red lines) compared with the same proteins with an opposite expression trend from the BL to the GS in the SSP group (blue lines). Heatmap of the functional annotation of the proteins are grouped. The red and blue boxes indicate the protein numbers, respectively. **e** Heatmap of protein expression (annotated with BM components and hemidesmosome complexes) between the normal and SSP groups. The right side of the heatmap shows the gene names. Red and blue boxes indicate the normalized intensities of proteins. The histogram shows the ratio of protein intensities from the BL and the SD in the SSP compared to the control groups. The Y-axis represents $\log_2$ SSP/control. **f** Immunofluorescence analysis of KRT14 and TGFBI (scale bar: 20 μm). The experiment was repeated three times. Source data are provided as a Source Data file.

3D EpSC cultures after XAV939 treatment, even after TGFBI was added (Fig. 5a), indicating that TGFBI could promote EpSC proliferation rather than EpSC differentiation. Next, we investigated whether the TGFBI-mediated regulation of EpSC growth occurred through Wnt signaling. The results showed that during EpSC proliferation, the expression levels of p-GSK3β and total β-catenin increased, and GSK3β and p-β-catenin (S33/37/T41) decreased, which could release β-catenin into the nucleus (Fig. 4k, Supplementary Fig. 7c). In addition, the expression of total β-catenin, cytoplasmic/nuclear fraction of β-catenin, and the transcription factor of Wnt signals (LEF-1) were increased (Fig. 4k–l), which activated the Wnt/β-catenin pathway with TGFBI treatment. Furthermore, the expression of cell–cell junction proteins, including tight junction (CLDN1), cadherin protein (ECAD), and desmosomes (DSC2, DSC3, and DSG1), was upregulated in a 3D epidermal organoid system with TGFBI treatment for one week (Fig. 5b–d).

Collectively, these results indicate that TGFBI, as an essential niche in the skin for basal EpSCs, can enhance the proliferation of EpSCs and maintain their stemness and cell junction involvement via the Wnt/β-catenin pathway.

**TGFBI enhanced skin regeneration through EpSC proliferation.** An in vivo mouse model of re-epithelialization was established to study the role of TGFBI-mediated EpSC proliferation in promoting skin regeneration. TGFBI was knocked down using siRNA interference, and the effects on wound repair were investigated (Supplementary Fig. 8a). The wound areas were measured three days, one week, and two weeks after the mice were treated with TGFBI siRNA. TGFBI knockdown significantly delayed wound healing (Fig. 6a, Supplementary Fig. 8b–c). The proliferation marker PCNA and EpSC marker KRT14 were downregulated following siTGFBI treatment for three days, one week, and two weeks, not only in the transitional regions but also at the edges of the epithelial wound (Fig. 6). In addition, the expression of the matrix-digesting proteases MMP1, MMP3, and MMP7, which are the secondary signals of cell migration, was downregulated to accommodate tissue expansion for proliferation (Fig. 6a, b, e, and Supplementary Fig. 8d). Wound healing was significantly enhanced following TGFBI treatment (Supplementary Fig. 8e), with significant increases in the expression of KRT14, PCNA, and the TGFBI interacted integrin ITGB1 (Supplementary Fig. 8f). These results suggest that TGFBI could promote skin re-epithelialization in vivo by enhancing the proliferation of EpSCs during wound healing. Next, the effect of TGFBI on SSP-derived EpSCs (SSP-EpSCs) was investigated.

**TGFBI enhanced the proliferation of SSP-EpSCs.** To determine whether TGFBI could restore the function of SSP-EpSCs, SSP-EpSCs were isolated and cultured for ten days (Fig. 7a). The stemness, proliferation, and cell–cell junction involvement of SSP-EpSCs decreased compared with those of the control group (Fig. 7b, c). TGFBI could rescue the differentiation process and improve the proliferation of SSP-EpSCs, and the expression levels of KRT14 and PCNA in EpSCs were restored to normal (Fig. 7b, c). The skin tissues in which SSP-EpSCs were present were prone to fibrosis during culture (Fig. 7d), and the expression level of the epithelial marker ECAD was deficient (Fig. 7c). The biological processes of apoptosis, immune response, and microbial invasion were upregulated in the SSP group compared with the control group. However, the processes were restored following TGFBI treatment (Fig. 7e and Supplementary Data 6). In particular, the biological processes of downregulated proteins in the SSP group included mitochondrial organization and translation, protein localization and posttranslational modification, DNA damage repair, and the cell cycle. Many proteins associated with the cell cycle, epidermal development, and nervous system development were downregulated in the SSP but restored following TGFBI treatment (Fig. 7f).

Taken together, these results indicate that TGFBI could enhance SSP-EpSC functions, such as proliferation, epithelialization, and epidermal development, and could be a potential therapeutic target for dermatologic injuries.

## Discussion

Throughout the life of an organism, the skin epithelium is constantly renewed and supplied by basal EpSCs, which are capable of proliferation and differentiation. The components of stem cell niches, such as secreted soluble factors and ECM, support self-renewal and facilitate tissue regeneration and repair by activating stem cell function[28]. Signals from both the epidermis and the dermis influence the rate of proliferation and behavior of EpSCs[29]. Therefore, EpSC-niche interactions are important for tissue development and fitness[30]. The BM, a thin layer of the dermal-epidermal junction, is derived from the epidermis and dermis and serves as the most critical structural and functional basis of skin tissue[3]. Usually, ECM proteins in the BM provide proliferative stimuli to the innermost BL of the epidermis via integrin receptors, such as α3β1 and α6β4[12]. Ablation of either one of these receptors in mice compromises BM assembly and impairs proliferation[31]. Therefore, a detailed analysis of the BM can shed light on the mechanism underlying skin homeostasis and wound repair.

Spirochetes primarily reside in the epidermis; however, one study of bacteria around the dermal capillaries was reported[32]. In

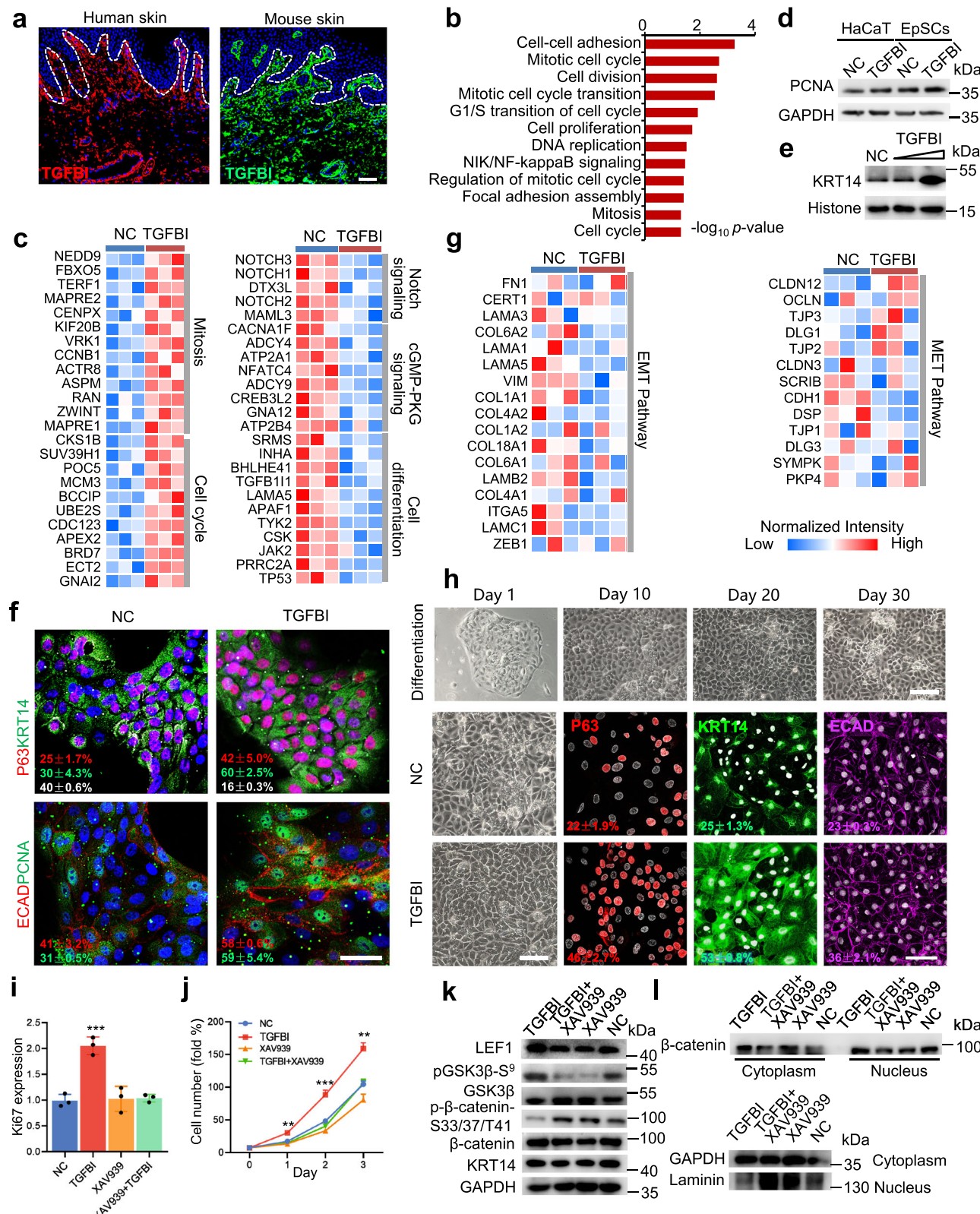

our previous study, spirochetes were detected in the epidermal tissues of all specimens ($n = 28$). In the present study, it was found that *T. pallidum* invasion of the epidermis severely affected the self-renewal ability and functional characteristics of EpSCs. These effects may cause an imbalance in EpSC proliferation, maintenance of stemness, epithelial characteristics, and cell junction-related ability, which could lead to EpSC fibrosis in the

skin tissues of SSP. Therefore, secondary syphilis can be an ideal model to study the function of EpSCs and their niche interactions. In this study, we used spatial proteomics and found that the expression of most BM-associated ECM proteins from BL and SD was downregulated in the skin tissues of SSP. In addition, the expression of TGFBI, a key glycoprotein that serves as a regulator of EpSCs and is found to be present in the BM zone, was

**Fig. 4 TGFBI enhanced the proliferation of EpSCs through the wnt/β-catenin pathway based on transcriptome analysis. a** Immunofluorescence of TGFBI in human and mouse skin tissues (scale bar: 50 μm). The white dashed lines represent the BM zone. **b** Biological process analysis of upregulated genes in the TGFBI-treated (100 ng/ml, $n = 3$) EpSCs of primary culture for 48 hours compared to the control group ($n = 3$). The genes were determined by the $p$ value (< 0.01) of a moderated $t$ test and $\log_2$ TGFBI/control > 1. **c** Functional analysis of differentially expressed genes (DEGs) between the TGFBI-treated ($n = 3$) and untreated ($n = 3$) samples based on transcriptome data. The DEGs were determined by the $p$ value of a moderated $t$ test. The exact $p$ values are provided in the Source Data file. **d** PCNA expression in TGFBI-treated HaCaT cells and EpSCs. **e** KRT14 expression in EpSCs cultured with TGFBI (50 and 100 ng/ml). **f** Immunofluorescence of p63, KRT14, KRT10, ECAD, and PCNA in TGFBI-treated EpSCs (scale bar: 50 μm). **g** Heatmap of EMT and MET pathway components identified in TGFBI-treated and untreated samples. The red and blue boxes in **c**, **g** indicate the normalized expression of the enriched or depleted genes, respectively. **h** Cell morphology and immunofluorescence of P63, KRT14, and ECAD in hiPSC-derived EpSCs cultured with TGFBI (scale bar: 50 μm). **i** Ki67 expression in TGFBI and XAV939-treated EpSCs for 48 hours. **j** Proliferation curve of TGFBI and XAV939-treated EpSCs from 0 to 72 hours. The data in **i**, **j** are shown as the mean ± SD ($n = 3$ per group). Significant differences between the TGFBI and other groups were determined by a two-tailed $t$ test (*$p < 0.05$, **$p < 0.01$, and ***$p < 0.001$). **k** Expression of LET1, GSK3β, pGSK3β-S[9], p-β-catenin-S33, total β-catenin, and KRT14 on TGFBI and XAV939-treated EpSCs. **l** Expression of cytoplasmic and nuclear fractions of β-catenin and laminin, and total GAPDH on TGFBI and XAV939-treated EpSCs. The experiments in **a**, **d–e**, **h**, and **k–l**, were repeated three times. Source data are provided as a Source Data file.

markedly downregulated in the epidermis and dermis of the SSP skin tissues.

TGFBI is a TGFβ-induced protein ig-h3 expressed in various tissues, including the bone, cartilage, heart, liver, and skin[33–36]. It has been reported that TGFBI modulates cell growth[37], tumorigenesis[38], wound healing[38], and apoptosis[33,39,40]. However, no relationship between TGFBI and stem cells has been reported. Despite some studies, which reported that TGFBI might participate in wound healing by playing a role in the adhesion and migration of keratinocytes/fibroblasts at wound sites[33,38,41], the regulation of TGFBI in wound healing has not been described in terms of stem cells. Here, we provided evidence for the regulation of TGFBI leading to enhanced EpSC proliferation through primary EpSCs, iPSC-derived EpSCs, and 3D epidermal organ models (Supplementary Fig. 9). Most TGFBI is synthesized and secreted by dermal fibroblasts and located in the BM zone to support EpSC stemness. This process occurs through the activation of the Wnt/β-catenin pathway, which plays a decisive role in skin homeostasis and wound repair[42]. During wound healing, EpSCs are activated and contribute to epidermal repair[43]. A full-thickness mouse dermatologic injury model was used to demonstrate that TGFBI could promote the re-epidermization process of wound repair by promoting the proliferation of EpSCs. SSP-EpSCs with dysfunctional stemness, epithelialization, proliferation, and skin development-associated signals were found to be restored following TGFBI treatment.

Collectively, our data suggest that the ECM plays an important role in regulating the growth of epidermal cells in the matrix microenvironment. The BM ECM is an essential component of the EpSC niche and can regulate the function of EpSCs.

## Methods

**Human subjects**. This study was approved in accordance with the ethical standards of the institutional board of the Peking Union Medical College Hospital with approval number ZS-2556. Written informed consent was obtained from all the patients and donors. The design and conduct of this study were in accordance with the Declaration of Helsinki.

Based on its infectious course, syphilis can be divided into the following main stages: primary, secondary, and tertiary[44]. Secondary syphilis is diagnosed according to the presence of a characteristic eruption, a reactive rapid plasma reagin test, particle agglutination assays for antibodies against *T. pallidum*, and/or fluorescent treponemal antibody absorption. Five male patients diagnosed with secondary syphilis between 2020 and 2021 were included, with ages ranging from 18 to 65 years. Clinical data, including the patients' age, sex, clinical features of the skin lesions, HIV status, and syphilis serological testing results, were collected (Supplementary Data 1). A skin biopsy was performed, and the tissue samples of SSP skin lesions were transferred to the laboratory for histopathological examination and epidermal stem cell (EpSC) culture within one hour. The samples were washed thoroughly with phosphate-buffered saline (PBS) containing gentamycin and amphotericin. Human foreskin tissues were obtained through circumcision as normal tissues. Four-micrometer-thick sections of each biopsy specimen were cut from formalin-fixed and paraffin-embedded tissue blocks. Following heat-induced epitope retrieval and primary rabbit polyclonal antibody

directed to *T. pallidum*, immunohistochemistry (IHC) staining was performed on a BenchMark ULTRA automated staining instrument (Roche Diagnostics) using an ultraView Universal Alkaline Phosphatase Red Detection Kit (Roche). Hematoxylin and eosin (H&E) staining results and IHC sections were reviewed by two expert dermatopathologists. Additionally, skin tissues from five male donors without syphilis were used as control groups, with ages ranging from 18 to 65 years.

**Preparation of human decellularized skin biomatrix scaffolds**. Decellularization was performed according to a previously described method[20,26]. Briefly, the skin tissues were rinsed three times with cold PBS containing 0.1% EDTA, followed by delipidation using phospholipase A2 combined with sodium deoxycholate for 4 h in a shaker at 37 °C or until the tissue segments became oyster white. Next, the skin surface was gently scraped using the back of a scalpel. The decellularized dermal samples (BM remained) were placed in sterilized 1.5-mL microcentrifuge tubes and rinsed at 37 °C for 1 h using 3.4 M NaCl for 1 h, followed by a final wash with PBS containing nucleases (10 μg/mL DNase, 5 μg/mL RNase). Finally, the decellularized dermal scaffolds were flash frozen for laser capture microdissection (LCM) and proteomics analysis.

**LCM of eight layers of skin samples**. LCM was performed using a laser microdissection system from Molecular Machines and Industries (MMI CellCut Laser Microdissection, Eching, Germany) that was controlled by the MMI Cell Tools software from the same company. The eight layers of skin samples were obtained according to the following procedure. First, five layers of native skin tissue were obtained (Supplementary Movies 1–5). The skin tissues were rinsed three times with cold PBS and frozen and embedded. Frozen 20-μm-thick sections of native skin tissues were cut and mounted on MMI MembraneSlides™ (MMI GmbH, Eching, Germany). After calibrating the laser focus and power, the stratum corneum, granulosum-spinosum, basal layer, superficial dermis, and deep dermis were cut successively. Finally, five layers of native human skin were obtained and attached to the EP tube cover. The second step was performed as follows: three layers of decellularized scaffolds were obtained (Supplementary Movies 6–8). Decellularized dermal scaffolds were obtained using the decellularization method (Supplementary Fig. 1). The scaffolds were frozen embedded, and frozen 20-μm-thick sections were cut from the scaffolds. After calibrating the laser focus and power, the basement membrane and the outermost part of the epidermis were cut, and the side of the decellularized scaffold was removed. Then, the superficial dermis and deep dermis were successively cut into EP tube covers.

**Sample preparation**. Samples from the five native skin tissue layers and three parts of decellularized skin scaffolds were scraped into a new EP tube with 20 μL of urea buffer (8 M urea, 150 mM Tris-HCl, 10 mM DTT, pH 8.0). An additional 10 μL of buffer was added to the tube. Steel balls were added to 30 μL of buffer for vibration (70 Hz) for 1 min. After centrifugation at $14,000 \times g$ for 10 min at 4 °C, the supernatants were transferred to clean tubes. Next, the extracted proteins were reduced at 37 °C for 1 h and alkylated in 25 mM iodoacetamide at room temperature for 30 min in the dark. The protein samples were digested with Lys C (1 μg at 37 °C for 4 h) and trypsin (enzyme-to-substrate ratio of 1:50 at 37 °C for 16 h), desalted through C18 cartridges (Beijing Qinglian Biotech, China) and vacuum-dried using a Speed Vac.

**Peptide prefractionation using high-pH HPLC**. The pooled peptides were fractioned using high-pH HPLC to reduce the sample complexity. Briefly, the peptides were dissolved in buffer A (2% acetonitrile [ACN], pH 9.5), loaded on an Xbridge C18 column (Waters, MA, USA; 4.6 mm × 100 mm, 130 A°, 5 μm), and eluted with a 70-min gradient from 0 to 95% buffer B (98% ACN, pH 9.5) at a flow rate of 0.7 mL/min. The aliquots were combined into 24 fractions before mass spectrometry (MS) analysis was performed.

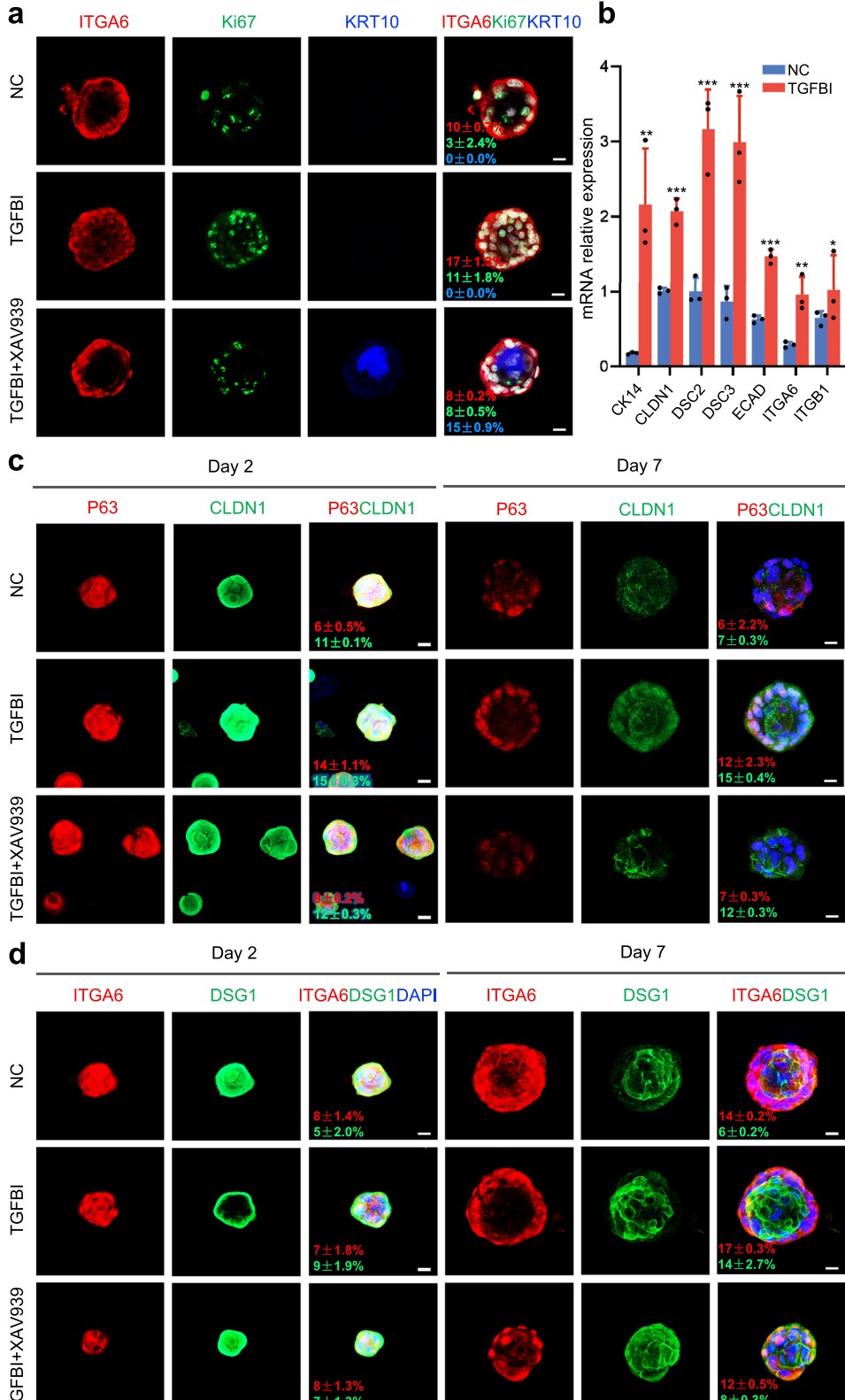

**Fig. 5 TGFBI enhanced epidermal organoid development. a** Immunofluorescence of ITGA6, Ki67, and KRT10 in epidermal organoids treated with TGFBI and XAV939 for seven days of culture (scale bar: 20 μm). The experiment was repeated three times. **b** Relative expression of EpSC marker and adhesion junction genes after TGFBI treatment. The data are shown as the mean ± SD ($n = 3$ per group). Significant differences between the TGFBI and control groups were determined by a two-tailed *t* test (*$p < 0.05$, **$p < 0.01$, and ***$p < 0.001$). Immunofluorescence of P63 and CLDN1 (**c**) and ITGA6 and DSG1 (**d**) in epidermal organoids treated with TGFBI and XAV939 for two and seven days of culture (scale bar: 20 μm). The experiment was repeated three times. Source data are provided as a Source Data file.

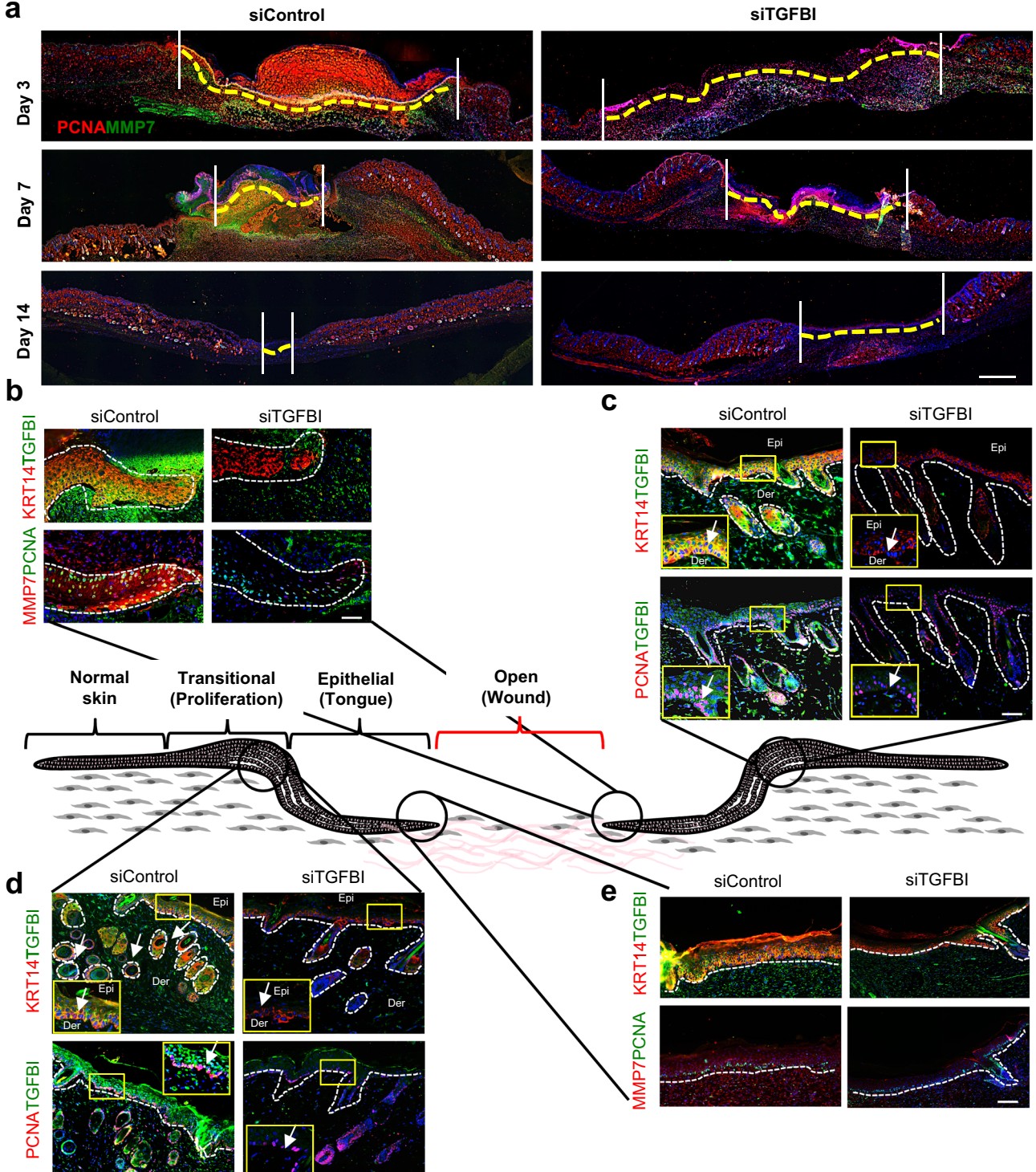

**Fig. 6 TGFBI promoted cutaneous wound healing through the enhancement of re-epithelialization. a** Immunofluorescence of wound healing on mouse skin under siTGFBI treatment at Days 0, 3, 7, and 14 (n = 3 per group, scale bar: 800 μm). Yellow dotted lines indicate unwounded skin areas. Immunofluorescence of MMP7, KRT14, TGFBI, and PCNA during the process of wound healing on mouse skin proliferation and tongue sites at Day 7 (**b**, **c**) and Day 14 (**d**, **e**) (scale bar: 50 μm).

**MS analysis**. The samples were measured using an EASY-nLC 1200 (Thermo Fisher Scientific, Waltham, MA, USA) coupled to a Q Exactive HF-X Orbitrap MS (Thermo Fisher Scientific) via a nanoelectrospray ion source (Thermo Fisher Scientific). Purified peptides were redissolved in mobile phase A (20% ACN and 0.1% formic acid) and directly loaded onto a C18 nanocapillary analytical column (Beijing Qinglian Biotech, China, 150 μm × 150 mm, 100 A°, 1.9 μm). For the proteome profiling samples, the peptides were separated on an analytical column over a 90-min gradient (buffer A: 0.1% formic acid and 80% H$_2$O; buffer B: 0.1%

formic acid and 20% ACN) at a constant flow rate of 0.6 μL/min (0–15 min, 8–12% buffer B; 15–65 min, 12–30% buffer B; 65–80 min, 30–40% buffer B; and 81–90 min, 95% buffer B).

The data-independent acquisition (DIA) scan mode was used for single-shot samples. The fractionated samples of the pool were acquired using the top 40 data-dependent acquisition (DDA) scan mode. Both acquisition schemes were combined with the same liquid chromatography gradient. The mass spectrometer was operated using Xcalibur software (version 4.1). The DDA scan settings on the full

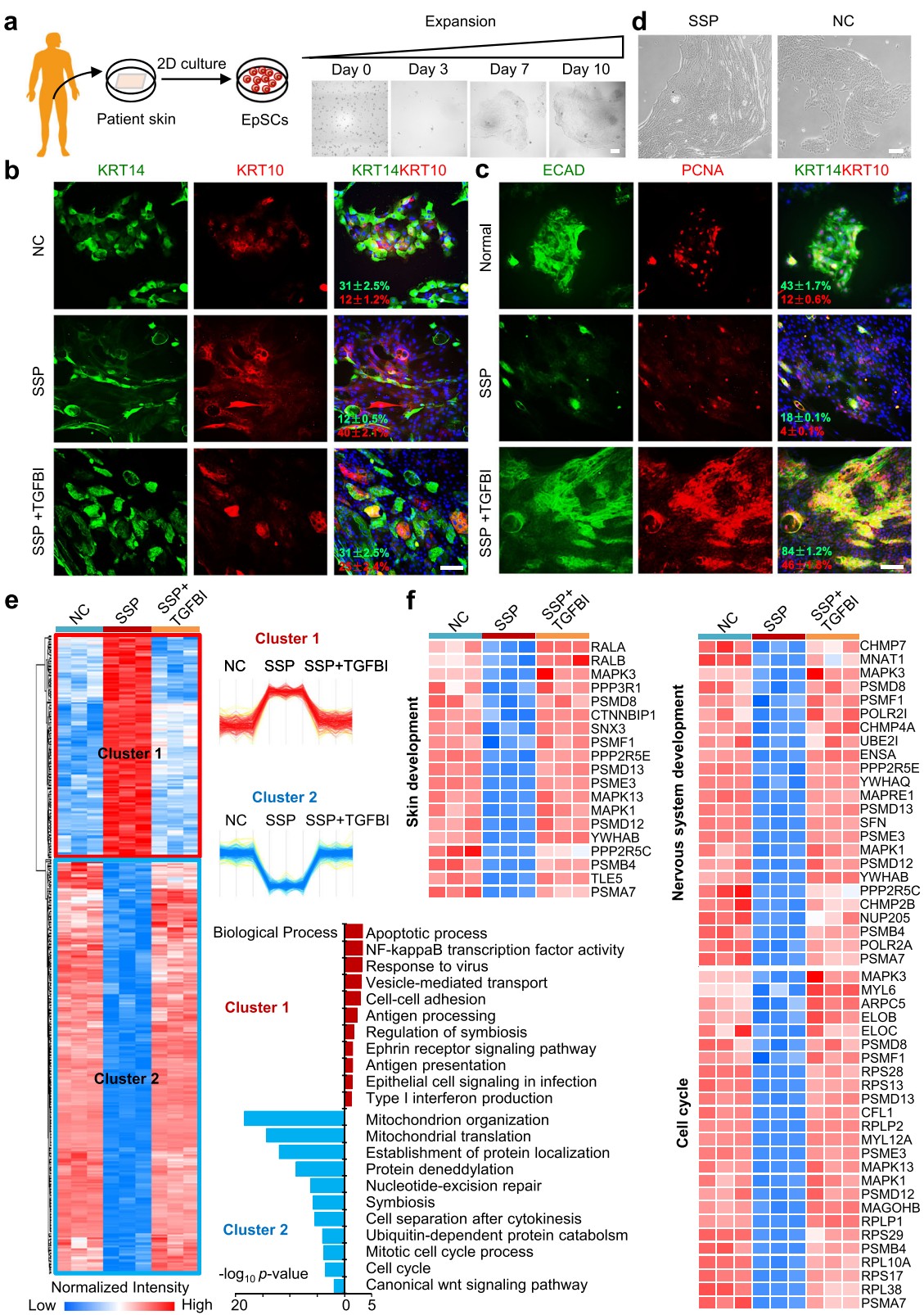

MS level included an ion target value of $3 \times 10^6$ charges in the 350–1,500 m/z range, with a maximum injection time of 80 ms and a resolution of 120,000 at 200 m/z. At the MS/MS level, the target value was $5 \times 10^4$ charges with a maximum injection time of 45 ms and a resolution of 15,000 at m/z 200. For the MS/MS events only, precursor ions with 2–7 charges not on the 16 s dynamic exclusion list were isolated within a 1.6 m/z window. Fragmentation was performed by higher-energy C-trap dissociation with a normalized collision energy of 27 eV. DIA was performed with one full MS event, followed by 42 MS/MS windows in one cycle. The full MS

settings included an ion target value of $3 \times 10^6$ charges in the 350–1,500 m/z range, with a maximum injection time of 50 ms and a resolution of 60,000 at m/z 200. DIA precursor windows ranged from 378 m/z (lower boundary of the first window) to 1345 m/z (upper boundary of the 42nd window). The MS/MS settings included an ion target value of $1 \times 10^6$ charges for the precursor window, with an Xcalibur-automated maximum injection time and a resolution of 30,000 at m/z 200.

For controlling the performance quality of the MS platform, the HEK293T (CRL-11268, Cell Resource Center, Peking Union Medical College) cell lysate was

**Fig. 7 TGFBI promoted the function of SSP-EpSCs. a** Schematic representation of the expansion of SSP-EpSCs (scale bar: 200 μm). Immunofluorescence staining of KRT14, KRT10 (**b**), ECAD, and PCNA (**c**) in SSP-EpSCs treated with TGFBI for 48 h (scale bar: 100 μm). The experiment was repeated three times. **d** Cell morphology of SSP-EpSCs after 14 days (scale bar: 200 μm). The experiment was repeated three times. **e** Heatmap analysis of the DEPs in SSP-EpSCs with or without TGFBI treatment in the control and SSP groups. The red and blue boxes indicate the normalized intensities of enriched or depleted proteins, respectively. The DEPs were determined by the Benjamini–Hochberg (BH)-adjusted $p$ value (<0.01) of a moderated $t$ test between SSP ($n = 3$) and control ($n = 3$) samples and SSP ($n = 3$) and TGFBI treatment ($n = 3$) samples. Cluster 1: $\log_2$ SSP/Control >0 and $\log_2$ (SSP + TGFBI)/ Control <−0; Cluster 2: $\log_2$ SSP/Control < 0 and $\log_2$ (SSP + TGFBI)/Control >−0. The biological processes of proteins in Clusters 1 and 2 are grouped according to the -$\log_{10}$ $p$ value of the degree of enrichment. **f** Functional analysis of the DEPs of Clusters 1 and 2. The columns on the left of the heatmap represent different functional categories. The right side of the heatmap shows the gene names. The red and blue boxes indicate the normalized intensities of enriched or depleted proteins, respectively. Source data are provided as a Source Data file.

measured as the quality-control standard sample every two days. The HEK293T standard sample was digested and analyzed using the same method, conditions, and MS instrument as for the skin samples. In the whole procedure of the MS experiment, six HEK293T standard samples were analyzed, and six MS datasets were generated. A pairwise Spearman correlation coefficient was calculated to perform protein quantification for all quality-control samples. The average correlation coefficient among the standards was 0.98, indicating the consistent stability of the MS platform.

**Proteomic MS/MS data processing**. The MS data of the fractionated pool (DDA data, 24 fractions) were analyzed by Proteome Discoverer (version 2.4). Then, the DDA data and the single-shot samples (DIA data) were used to generate a DDA library and direct DIA library, respectively, which were computationally merged into a hybrid library in Spectronaut (version 14.9.201124.47784, Biognosys, Switzerland). Then, the raw DIA data were processed on Spectronaut using the default settings. All searches were performed against the human UniProt reference proteome sequences, with 20,279 entries that were downloaded in February 2021. The searches used carbamidomethylation as the fixed modification and acetylation of the protein N-terminus and oxidation of methionines as the variable modifications. Default settings were used for other parameters. In brief, a trypsin/P proteolytic cleavage rule was used, permitting a maximum of two miscleavages and a peptide length of 7–52 amino acids. Protein intensities were normalized using the "Local Normalization" algorithm in Spectronaut based on a local regression model, and the retention time prediction type was set to dynamic iRT and correction factor for a window. The mass calibration was set to local mass calibration. Decoy generation was set to Inverse. Interference correction on the MS2 level was enabled, removing fragments for quantification based on the presence of interfering signals, but at least three fragments were maintained for quantification. The false discovery rate was estimated using the mProphet approach and set to 1% at the peptide level.

**Human EpSC culture**. EpSCs were cultured using previously described methods[45]. Briefly, following digestion with 2 mg/mL Dispase II at 37 °C for 1 h, the epidermal sheets were separated from the dermis and were digested with prewarmed (37 °C) 0.25% trypsin/EDTA for 15 min. The digestion was stopped with fetal bovine serum, and the epidermal cells were washed twice with cold PBS to remove trypsin, filtered through a 40-mm strainer and centrifuged at 1000 rpm for 3 min. For 2D (monolayer) cultures, the cells were seeded onto plates precoated with Matrigel. Advanced DMED/F-12 medium supplemented with 1% NEAA, 2% B-27, 1% Glamax, 1% HEPES, 1.25 mM N-ace, 50 ng/mL hEGF, 50 ng/mL Wnt3a, 1 μM A83, 10 μM Fosklin, and 10 μL/mL 1× penicillin/streptomycin was added. The EpSCs of healthy individuals and patients with secondary syphilis were treated with TGFBI for 48 h. For 3D cultures (organoids), the isolated epithelial cells of human foreskins were cultured on ultralow-attachment culture dishes in the medium mentioned above with 0.2% sodium hyaluronate, with or without TGFBI. After 24 h, all the culture medium in the wells was removed and replaced with fresh medium. After 8 days of culture, 3D organoids were harvested for the next experiment.

**Culture of hiPSC-derived EpSCs**. To differentiate the EpSCs, the human iPSC line nciPS02 (hiPSC, RC01001-B, Female, Nuwacell Biotechnologies Co., Ltd) was seeded onto plates precoated with Matrigel and maintained in mTeSRTM1 medium until a confluency of 90% was attained. For differentiation, the hiPSCs were incubated in defined keratinocyte serum-free medium (KSFM) supplemented with 1 μM retinoic acid, 10 ng/mL bone morphogenetic protein 4, and 1% penicillin–streptomycin, as previously described[46]. Every 2 days, 50% of the culture medium was removed and replaced with fresh medium. After 30 days, the hiPSC-derived EpSCs were incubated in KSFM supplemented with 100 ng/mL TGFBI recombinant protein for 48 h. The characterization of hiPSC differentiation was performed on Days 10, 20, and 30 using imaging.

**Mouse wound healing experiment**. All animal experiments were performed following standards with the approval of the Institutional Animal Care and Use Committee at the National Center for Protein Sciences (Beijing). A total of 21

C57BL/6 wild-type male mice (6–8 weeks) were purchased from Beijing Vital River Laboratory Animal Technology Co., Ltd. (Beijing, China). The mice were housed in a temperature-controlled room under a 12-h light/12-h dark cycle under pathogen-free conditions. All male mice were anesthetized using ketamine, and the mouse hair on the back was shaved using an electric shaver. Then, depilatory cream was applied evenly on the backs for 30 s to remove the hair before surgery. For the TGFBI knockdown experiments, the following two groups of mice were used: (1) control siRNA-treated and (2) TGFBI siRNA-treated. Each group comprised nine mice, three of which were used per time point. After disinfection with povidone-iodine, a 10-mm-diameter circle of the epidermis in the back was removed. The wound sites were sterilized after tissue removal. The wounded skin was smeared with 5 nmol of chemically-modified siRNA of TGFBI or control siRNA once every 24 h. A silica gel circle and a 3-mm wound plaster were firmly laid over the wounded skin using a sterilized gauze bandage. Each mouse was marked and placed into a rearing cage. Images of the wounded skin were taken at 0 days, 3 days, 7 days, and 14 days postoperatively, and wounded tissues were obtained from the mice ($n = 3$ for each group). A dose of 2 μg TGFBI recombinant protein with 10% DMSO was applied to one wound site twice a day, and 10% DMSO was applied to the other side as a control. The mice were sacrificed 3 days, 5 days, and 1 week after treatment ($n = 3$ for each group). The wound tissues were fixed in 4% formaldehyde for immunochemical staining and analysis.

**H&E staining, histology, and immunofluorescence**. The samples were fixed in 4% formaldehyde overnight at 4 °C and then processed through a dehydration gradient. After being embedded in paraffin, the tissues were cut into 4-μm-thick sections for H&E staining, immunohistochemistry, and immunofluorescence analysis. For immunohistochemical staining, the sections were deparaffinized and heated in a microwave to a boil for at least 12 min with antigen retrieval buffer and then removed by endogenous catalase in 0.3% $H_2O_2$ for 30 min after cooling. The sections were then blocked with normal horse serum in Tris-buffered saline for 1 h, blocked with an Avidin/Biotin Blocking Kit, and stained with antibodies overnight at 4 °C. The sections were then stained with a secondary antibody. For immunofluorescence staining, the cells were fixed in 4% formaldehyde for 20 min and then washed in PBS. Then, the cells were treated with 0.25% Triton X-100 for 20 min to remove plasma membranes, blocked in 10% serum for 1 h at room temperature (RT), and incubated with primary antibodies overnight at 4 °C. After incubation for 1 h at RT with secondary antibodies and counterstaining with DAPI, the sections were sealed with Fluoro-Gel for photography. Negative control samples were incubated with secondary antibodies alone. Images were taken at 20×/40× magnification and analyzed using Volocity Demo (×64).

**Confocal microscopy**. Immunofluorescence imaging was used to observe the polarity, cell junctions, proliferation, and differentiation markers of EpSCs and hiPSCs-EpSCs. The cells were cultured in glass-bottomed dishes, fixed in 4% formaldehyde for 20 min, and then washed in PBS. The samples were then treated with 0.5% Triton X-100 for 15 min and subsequently blocked for 1 h with 10% serum at RT. Then, the samples were incubated with primary antibodies overnight at 4 °C. The samples were then incubated for 1 h at RT with secondary antibodies and counterstained with DAPI. Images were taken using an LSM 700-point scanning confocal microscope equipped with 5× and 10× objectives.

The protein expression level in the immunofluorescence staining experiment was measured by the intensity of fluorescence. ImageJ software (version 1.53) was used to detect the fluorescence intensities and quantify the protein expression. For each single channel (monochrome) fluorescence picture, the gray value of each pixel represents the fluorescence intensity of the point. The fluorescence intensity of a specific region was calculated as follows: average fluorescence intensity (Mean) % = sum of fluorescence intensity of the region (IntDen)/Area of the region (Area) *100%.

**Western blot analysis**. EpSCs and hiPSC-EpSCs were solubilized in 1 mL of cold lysis buffer (50 mM Tris-HCl pH 8.0, 150 mM NaCl, 1% Triton X-100) containing complete protease inhibitors and shaken in an ultrasonic cell disruptor at 4 °C until the buffer solution was clear. The samples were extracted in SDS sample buffer and used for immunoblotting. The antibodies were incubated for 2 h at 37 °C. After

being washed three times, the blots were incubated with secondary antibodies for 50 min at 37 °C. After the blots were washed another three times, signals were detected using Pierce™ ECL Western blotting Substrate and Millipore Immobilon™ Western Chemiluminescent HRP Substrate at room temperature.

**QPCR**. According to the manufacturer's instructions, total RNA was isolated using an RNeasy Mini Kit; then, cDNA was synthesized using reverse transcriptase. qRT–PCR was performed with SYBR green master mix on a Bio-Rad iQ5 Real-Time PCR detection system. Data were collected using Bio-Rad CFX Manager software, and the expression of genes within a sample was normalized to GAPDH expression using the $2^{-\Delta\Delta Ct}$ method. The PCR primer sequences are shown in Supplementary Data 7.

**Statistical and bioinformatics analysis**. General data analysis was performed using R package software (version 4.0.3). The quantification values of the identified proteins were normalized by taking the fraction of the total, followed by multiplication by $10^6$ and log2 transformation. Pairwise comparisons to identify proteins in which the expression was significantly different between patients with syphilis and the controls were performed by a moderated t-statistic using the R package Limma (version 3.46.0)[47]. Differences for which the Benjamini–Hochberg adjusted $p$ value were <0.01 were considered statistically significant. For the other experimental results, the statistical data are shown as the means ± standard deviation (SD). Student's $t$ test was performed to compare data between groups. ANOVA was used for comparisons between three or more groups. Each experiment was performed at least three times ($n ≥ 3$). The results were considered significant at $p ≤ 0.05$ (*), 0.01 (**), or 0.001 (***). Statistical tests were conducted using GraphPad Prism (version 9.0.0).

The online tool DAVID (https://david.ncifcrf.gov/)[48] was used to annotate the proteins according to biological processes and cellular components via Gene Ontology[49] analysis, and the enrichment of biological pathways in differentially expressed proteins was assessed using Kyoto Encyclopedia of Genes and Genomes[50] pathway analysis (https://www.kegg.jp/kegg/pathway.html). The protein interactome network for TGFBI was constructed using Cytoscape (version 3.8.2)[51], and the protein–protein interactions were retrieved from the STRING database[52]. Principal coordinates analysis of proteins in which the values in each sample were valid was performed using the R package ape (version 5.4.1)[53]. The heatmaps of protein quantitation were displayed using Perseus software (version 1.6.0.7)[54]. The Matrisome database[55] (http://matrisomeproject.mit.edu/proteins/) was used for ECM annotation to define the core ECM proteins (including collagens, proteoglycans, and ECM glycoproteins) and ECM-associated proteins (including ECM regulators, ECM-affiliated proteins, and secreted factors).

**Reporting summary**. Further information on research design is available in the Nature Research Reporting Summary linked to this article.

## Data availability

The proteomics data generated in this study have been deposited to the ProteomeXchange Consortium (http://proteomecentral.proteomexchange.org/) via the iProX[56] partner repository (https://www.iprox.cn/) under accession code PXD027093. All equipments, reagents, and supplies are available in the Supplementary Data 8. Source data are provided with this paper.

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

## Acknowledgements

We thank Beijing Qinglian Biotech for the technical support of mass spectrometry identification experiment. This work was supported by Beijing Municipal Science and Technology Commission (Z191100006619011), Capital's Funds for Health Improvement and Research (2020-2-4016), CAMS Innovation Fund for Medical Sciences (CIFMS 2020-I2M-C&T-B-048), and National Science and Technology Major Project (2021YFA1301603).

## Author contributions

L.L., J.L., J.M., and Y.Z. conceived the overall study and designed experiments. L.L., J.L., and Y.Z. had full access to all the data in the study and take responsibility for the data analysis accuracy. L.L., J.M., Y.Z., and X.L. performed proteomics experiments and bioinformatics analysis. J.L., H.Z., and Z.W. performed the skin tissues and ethics preparation. J.L., Q.Z., H.G., and Y.W. performed laser capture microdissection experiments. Q.Z. and D.G. participated in cell culture experiments. Q.Z., B.L., and D.G. performed the mouse wound healing experiment. L.L., J.M., J.L., and Y.Z. wrote and edited the manuscript. All authors made important comments on the manuscript.

## Competing interests

The authors declare no competing interests.
