## [Peer Review File · Nature Communications]

Spatially resolved proteomic map shows that extracellular matrix regulates epidermal growthEditorial Note: Parts of this Peer Review File have been redacted as indicated to remove third-party material where no permission to publish could be obtained.

REVIEWER COMMENTS

Reviewer #1 (Remarks to the Author):

In this manuscript, authors used laser capture microdissection and mass spectrometry to construct the spatial proteome of stratified human skin. By comparing normal skin to the skin tissues with secondary syphilis (SSP), they identified TGFBI as a key factor that enhances the growth of epidermal stem cells (EpSC) and the process of wound healing. This manuscript provides a vast amount of proteomic data which would be useful for future research; however, unclear interpretation and data presentation lead to inadequate conclusion and would cause misleading to the readers. Although their data are potentially interesting, appropriate editing on their interpretation, English writing and additional data analysis are essential before it can be recommended for publication. Several concerns are listed as follows:

1. Overall, the presentation of figures should be re-organized for a better appreciation; e.g Fig. 3, sub-figures are not arranged in order. Fig.S6k was mentioned in the main text before Fig.6i, j; thus, the sequence should be switched accordingly. Moreover, quantification data for several immunostaining analyses are missing, e.g. Fig.4g, 4k, 4o, 5a, 5c, 5d and 7c.
2. In addition, although authors tended to link TGFBI to Wnt signaling activation, the information about how TGFBI activates Wnt signaling is very limited. The effects of Wnt signaling inhibitor, XAV939, on cell proliferation and protein expression could be independent of TGFBI. Authors will need to show direct evidence to demonstrate that TGFBI-mediated regulation on EpSCs depends on Wnt signaling.
3. Fig.3f & Fig.4a: The immunostaining for TGFBI are so different between these two figures. In Fig.3f, TGFBI seems highly expressed at BM and some in basal layer of epidermis; in Fig. 4a, TGFBI staining is exclusively located in dermis. Please explain this inconsistency.
4. Fig.4l: In this western blot authors should also show (1) total GSK3 β , (2) the levels of p- β -catenin (S33/37/T41), p- β -catenin (S552) and total β -catenin, and (3) cytoplasmic / nuclear fraction of β -catenin. Without knowing the status of β -catenin, elevated p-GSK3 β (S9) is not sufficient to demonstrate the activation of Wnt/ β -catenin signaling.
5. Fig.4m & 4n: The data sets from XAV939-treated cells are missing. In addition to measuring Ki67 mRNA expression, cell proliferation should be determined either by counting cell numbers over the course of time or by performing MTT assays.
6. Fig.4o: β -catenin staining on EpSC treated with XAV939 and XAV939+TGFBI should be shown in order to demonstrate the effect of TGFBI on β -catenin signaling activity. Nuclear β -catenin needs to be quantified accordingly.
7. Fig.5: As not all the images are in the same focal plane, the presented images between samples are not comparable. Using confocal microscopy is necessary to show the expression levels and localization of indicated proteins in the cells at the surface of the organoid and inside organoid. "Quantification" of immunostaining data is required.
8. Fig.5c: Some of immunofluorescence staining images are over-exposed, e.g. Fig.5c middle panel for TGFBI. Also, CLDN1 seems not located at cell junction in the cells treated with TGFBI for 7 days.
9. Fig.6a: The proof for TGFBI knockdown should be provided, and the delayed wound healing upon

TGFBI knockdown should be evidenced by the quantification data of wound closure.

10. Fig.6e: According to the morphology, the tissue sections shown in Fig.6e seem not located at epithelial tongue. Please confirm it. There is a mislabel in the lower panel: KRT14 should be PCNA.

11. Fig.7d & Line 379: "The skin tissues in which SSP-EpSCs were present were prone to fibrosis during culture." Cutaneous fibrosis, the result of aberrant process leading to abnormal deposition of ECM in the dermis, is commonly used to describe the underlying dermis, not epidermal cells. Did authors mean to say "EpSCs undergo epithelial-mesenchymal transition"? or did authors identify "excess production of ECM proteins by SSP-EpSCs"? Please explain it.

12. Fig.S4g: "Different shapes represent different signaling pathways related to the interacting proteins." No different shapes were presented in this interactome network. Could authors also define which proteins are involved in Wnt signaling?

13. Fig.S6e: Two bands are shown for TGFBI in medium fraction. Please explain it.

14. Line 83-84: there are several phosphorylation sites for β -catenin; some are for degradation, e.g. S45, S33/37/T41; some are for activity and nucleus accumulation, e.g. S552. It should be defined clearly. The references should be updated to more recent articles.

15. Line 290: "...syphilis infection destroyed epidermal development and...". Epidermal development usually indicates the developmental process during embryogenesis. It would be more appropriate to use "epidermal homeostasis" or "epidermal stratification".

16. Line 343-350: No description or data interpretation on the effects of XAV939 in TGFBI-treated organoids. It remains unclear the relationship between Wnt signaling and TGFBI.

17. Line 363-364: Aren't MMP proteins involved in cell migration rather than cell proliferation? Please confirm it.

18. English writing needs to be edited carefully. For examples:

- Line 313: "TGFBI expression could be induced with TGF β 1 and was secreted in the extracellular space in fibroblasts." should be "TGFBI expression could be induced by the treatment of TGF β 1 and was secreted in the extracellular space in between fibroblasts".

- Line 330: According to the diagram shown in Fig.4j, "EpSC-derived hiPSCs (EpSC-hiPSCs)" is not an accurate term; it should be "EpSCs derived from hiPSCs" or "hiPSC-derived EpSCs".

Reviewer #2 (Remarks to the Author):

I was asked to evaluate the proteomics portion of this work. The authors did an excellent job of designing the study, from the information I had available to me. All the standard tools were used for processing and analysis, and this work was done at a very high level. The authors are to be commended for doing replicate injections of replicate samples for each skin layer; that gives the absolute best/most reliable quantification in this type of experiment.

If the authors truly did not randomize their samples at all, though, that is somewhat problematic. Under these circumstances, it is difficult to conclude whether any significant changes are due to real, biological differences or due to instrument drift over time or other possible confounding effects. Also, I was not able to examine the raw data in the ProteomeXchange submission because I did not have the login information.

From the information available to me, this work appears to be very good. However, I would like to examine the data in ProteomeXchange and also have the authors address the issue of sample randomization.

Reviewer #3 (Remarks to the Author):

This data-rich and generally well-written manuscript describes use of tissue-specific proteome and transcriptome analysis. In general, the data are convincing. However, the results are incompletely described, and in particular the figure legends contain insufficient information for the reader to be able to interpret them. Pertinent prior studies are not incorporated in the interpretation or cited. The overall meaning of the results are also not described adequately.

The manuscript does not adequately cite or integrate prior findings. See, for example, Cruz et al., *PLoS Negl Trop Dis.* 2012;6(7):e1717. doi: 10.1371/journal.pntd.0001717. Epub 2012 Jul 17.

1. l. 179-181 and elsewhere. In many places in the manuscript, varied abundance of proteins in different tissue regions is referred to as upregulation or downregulation. This terminology is incorrect, because (as exemplified by keratins in the epidermis and collagen I in the dermis) many proteins accumulate during development. Their levels do not necessarily correlate with the relative mRNA levels or ONGOING protein production in the tissue region. For example, the stratum corneum lacks nuclei and likely has little new mRNA and protein production; most of the proteins present was produced during the cells' development in the stratum spinosum and granulosum. The proteins are also subject to differential degradation and modification. Therefore it would more appropriate to refer to differences in levels of protein abundance rather than upregulation or downregulation.

2. l. 197. Does it make sense that proteins associated with neuronal development are present in high levels in the GS layer? It would be better to examine the underlying functions of the proteins, which in this tissue would be expected to have little role in neuronal development.

l. 362 and Fig. 6. The use of the term 'epithelial tongue' would be confusing to most readers. Alternative wording should be used.

l. 486. The equipment used for laser dissection is not described.

The manuscript would benefit from a figure showing a model of how TGFBI expression is proposed to be related to epidermal development, EpSC proliferation, and expression of the proteins involved in differentiation of the layers. As it stands, the overall findings are unclear.

In general, the figure legends should be sufficiently detailed so that the reader does not have to refer to the body of the manuscript to easily understand the figure. That is not the case for any of the figures. The figures would also benefit from labeling and arrows showing key features. Specific comments are provided below.

Fig. 1. Further labeling and explanation are needed in the figure and legend. The first sentence should state "in normal human skin" or something similar. The images in 1a should have labeling for the epidermis, basement membrane and dermis, and the legend should describe the coloration of each specific label and the DAPI staining. In 1c, "Spatially" should be changed to "Spatial", and the validation description should be expanded somewhat. In 1d legend, "Number of proteins detected" is suggested. In 1g, the meaning of "summed intensities of the proteins of interest by the summed intensities of all proteins" is unclear, in that the summed values for each region do not approach 100 (particularly for the basement membrane).

Fig. 2. For the 2a legend, it should be indicated that each column within the skin regions corresponds to a different human skin specimen (if that is the case). Why is the number variable? Perhaps dashed lines could be drawn at the upper and lower boundaries to more clearly associate the regions of the heat map with the descriptions on the right side. In 2c, the EGFR and COL4AJ specimens appear to be IHC rather than immunofluorescence, and the color difference between the specific staining and nonspecific stain (hematoxylin?) is not sufficient to discern. These should be redone. As for 1a, 2c should be more thoroughly described in the legend; perhaps arrows could be added to indicate areas of specific staining.

Fig. 3. In 3a, what is the staining shown under "Treponema pallidum"? The staining method is not described. Also, the coloration in the lower right panel of 3a does not match that of the low magnification view. Scale bars should be used consistently and labeled. In 3b legend, SSP-GS should be described as the stratum granulosum-spinosum layers rather than stratum corneum. For 3d legend, the term "gradually" does not fit here. In 3e, the NC and SSP groups should be more clearly separated in the figure.

Fig. 4. The title sentence should indicate that the data in this figure is based primarily on RNA-seq analysis, as opposed to protein analysis in the preceding figures. In 4a, the coloration and the meaning of the dashed line should be described. In 4b legend, it should be stated that these are in vitro cultures of EpSCs. The treatment interval (hours) should be indicated. For 4h, the x axis should be labeled, and a more standard y axis (rather than e5) should be used. Fig. 4j is unclear and should be omitted.

l. 248 and elsewhere. expression rather than expressions.

REVIEWER COMMENTS

Reviewer #1 (Remarks to the Author):

In this manuscript, authors used laser capture microdissection and mass spectrometry to construct the spatial proteome of stratified human skin. By comparing normal skin to the skin tissues with secondary syphilis (SSP), they identified TGFBI as a key factor that enhances the growth of epidermal stem cells (EpSC) and the process of wound healing. This manuscript provides a vast amount of proteomic data which would be useful for future research; however, unclear interpretation and data presentation lead to inadequate conclusion and would cause misleading to the readers. Although their data are potentially interesting, appropriate editing on their interpretation, English writing and additional data analysis are essential before it can be recommended for publication. Several concerns are listed as follows:

1. Overall, the presentation of figures should be re-organized for a better appreciation; e.g Fig. 3, sub-figures are not arranged in order. Fig.S6k was mentioned in the main text before Fig.6i, j; thus, the sequence should be switched accordingly. Moreover, quantification data for several immunostaining analyses are missing, e.g. Fig.4g, 4k, 4o, 5a, 5c, 5d and 7c.

Answer: Many thanks for the editor's suggestion.

In the revised manuscript, we have re-organized Fig. 3 and Fig. S6 to make the sub-figures in order. Also, we have added the quantified data for immunostaining analyses in Fig 4f, 4h, 5a, 5c, 5d, 7b and 7c.

Fig. 3 in the revised manuscript.

Fig. S6 in the revised manuscript.

Fig. 4f, h in the revised manuscript.

Fig. 5 in the revised manuscript.

Fig. 7b, c in the revised manuscript.

2. In addition, although authors tended to link TGFBI to Wnt signaling activation, the information about how TGFBI activates Wnt signaling is very limited. The effects of Wnt signaling inhibitor, XAV939, on cell proliferation and protein expression could be independent of TGFBI. Authors will need to show direct evidence to demonstrate that TGFBI-mediated regulation on EpSCs depends on Wnt signaling.

Answer: Many thanks for the reviewer's suggestion.

In the revised version, we performed experiments to provide direct evidence and the explanation for the correlation between TGFBI and Wnt signaling on EpSCs.

First, we found clues through the interaction between TGFBI and proteins identified in BL. As shown in Fig. S4f, TGFBI interacted with several Wnt pathway related proteins, including ITGB1 and ITGA3. We also proved that the expression of ITGB1 increased after TGFBI treatment for EpSCs (Fig. 5b). Then, we found that with TGFBI treatment, the expression of transcription factor (LEF-1) of Wnt signal pathway increased (Fig. 4k). Besides, total β -catenin expression and its expression in nucleus increased (Fig. 4l). It is known that a small part of β -catenin is phosphorylated (such as S33/37/T41) under natural conditions (*Cell*. 2002, 108(6):837-47.). Phosphorylated β -catenin and GSK3 β can form complexes and be degraded by ubiquitination. Our results showed that after TGFBI treatment, the total GSK3 β expression decreased, phosphorylated GSK3 β increased, which may lead to a decrease of β -catenin degradation (Fig. 4k and Fig. S6i). When XAV939, which can promote β -catenin degradation, was added in the EpSCs, even with the treatment of TGFBI, the degradation of β -catenin was not improved (Fig. 4k, l and Fig. S6h). In addition, with XAV939 and TGFBI treatment, the expression of total GSK3 β did not decrease, phosphorylated β -catenin (S33/37/T41) increased (Fig. 4k), and the expression of β -catenin in the nucleus decreased (Fig. 4l). These results showed the regulatory relationship between TGFBI and Wnt pathway.

On the other hand, we found that the proliferation and stemness associated proteins of EpSCs increased after TGFBI treatment. When we added siTGFBI, the proliferation and stemness ability

of EpSCs decreased slightly in EpSCs, while the expression level of EpSCs function related proteins (PCNA and total β -catenin) recovered after TGFBI treatment (Fig. S6j). These results indicated that TGFBI can regulate EpSCs' function. After adding XAV939 or adding TGFBI after adding XAV939, we found that while Wnt signals were not activated, the function associated markers of EpSCs could not be restored. The above results provide the evidence for TGFBI-mediated regulation on EpSCs depends on Wnt signaling.

Fig. S4f in the revised manuscript.

Fig. 5b in the revised manuscript.

Fig. 4i-l in the revised manuscript.

Fig. S6h-j in the revised manuscript.

3. Fig.3f & Fig.4a: The immunostaining for TGFBI are so different between these two figures. In Fig.3f, TGFBI seems highly expressed at BM and some in basal layer of epidermis; in Fig. 4a, TGFBI staining is exclusively located in dermis. Please explain this inconsistency.

Answer: Many thanks for the reviewer's suggestion.

According to our previous experiments, TGFBI is expressed in dermal fibroblasts after TGFβ stimulation, while TGFBI is almost not expressed in epidermal cells (Figure 1). Therefore, there should be a large amount of TGFBI in the dermis. In this study, the TGFBI protein has been identified with a high intensity in basement membrane by LC-MS/MS method; further, through the immunochemical staining verification, TGFBI is proved located in the basement membrane (Fig. 4a and Fig. S6a). It is speculated that the TGFBI protein is secreted by fibroblasts first and then located in the basement membrane. As the structure of pathological section varies with different tissue part, the expression level of TGFBI in different sections may differ slightly. To make the results more clear, we have updated the staining pictures in Fig. 3f. As shown in Fig. 3f and Fig. 4a, there is a large amount of TGFBI in the basement membrane (dotted line) and superficial dermis of skin tissues. In addition, the dermis contains blood vessels and TGFBI also locates in basement membrane under the vascular endothelial cells. Therefore, it seems that more TGFBI is expressed on the pathological section of dermis with more blood vessels.

Figure 1.

Fig. 4a in the revised manuscript.

Fig. 6a in the revised manuscript.

Fig. 3f in the revised manuscript.

4. Fig.4l: In this western blot authors should also show (1) total GSK3 β , (2) the levels of p- β -catenin (S33/37/T41), p- β -catenin (S552) and total β -catenin, and (3) cytoplasmic / nuclear fraction of β -catenin. Without knowing the status of β -catenin, elevated p-GSK3 β (S9) is not sufficient to demonstrate the activation of Wnt/ β -catenin signaling.

Answer: Many thanks for the reviewer’s suggestion.

In order to demonstrate the activation of Wnt/ β -catenin, we added the western blot results of total GSK3 β and the levels of p- β -catenin (S33/37/T41) in Fig. 4k. The expression of total β -catenin have been provided in Fig. 4k; and the levels of cytoplasmic / nuclear fraction of β -catenin have been provided in Fig. 4l. In addition, we added the results of the increased expression of transcription factor (LEF-1) of Wnt signal pathway after TGFBI treatment (Fig.4k). Further, we found that knockdown of TGFBI can lead to the decrease of β -catenin; while knockdown of TGFBI with TGFBI supplement, the β -catenin restore to the normal level (Fig. S6j). These results indicated the regulated relationship between TGFBI and Wnt/ β -catenin signaling.

Fig. 4k, l in the revised manuscript.

Fig. S6j in the revised manuscript.

5. Fig.4m & 4n: The data sets from XAV939-treated cells are missing. In addition to measuring Ki67 mRNA expression, cell proliferation should be determined either by counting cell numbers over the course of time or by performing MTT assays.

Answer: Many thanks for the reviewer's suggestion.

The data of XAV939-treated cells has been added in Fig.4i and 4j (refer to Fig. 4m and 4n in previous version). Also, the result of cell number changes (fold) over the course of time has been provided with growth curve in Fig. 4j

Fig. 4i, j in the revised manuscript.

6. Fig.4o: β-catenin staining on EpSC treated with XAV939 and XAV939+TGFB1 should be shown in order to demonstrate the effect of TGFB1 on β-catenin signaling activity. Nuclear β-catenin needs to be quantified accordingly.

Answer: Many thanks for the reviewer's suggestion.

The results of β-catenin staining on EpSC treated with of XAV939 and XAV939+TGFB1 have been added in Fig. S6h. The WB experiment has been used to quantify the staining results of nuclear and cytoplasm β-catenin and the results have been added in Fig. S4l.

Fig. S6h in the revised manuscript.

Fig. 4l in the revised manuscript.

7. Fig.5: As not all the images are in the same focal plane, the presented images between samples are not comparable. Using confocal microscopy is necessary to show the expression levels and localization of indicated proteins in the cells at the surface of the organoid and inside organoid. “Quantification” of immunostaining data is required.

Answer: Many thanks for the reviewer’s suggestion.

These images are the staining results of 3D spheroids synthesized by different layers including the inside layers of organoids. The following Figure 2 shows the images of the 3D spheroids at different steps of the z-axis, as example to show the expressions of proteins in the organoids. Besides, the quantitative results of fluorescent staining have been added with Fig. 5.

Figure 2.

8. Fig.5c: Some of immunofluorescence staining images are over-exposed, e.g. Fig.5c middle panel for TGFBI. Also, CLDN1 seems not located at cell junction in the cells treated with TGFBI for 7 days.

Answer: Many thanks for the reviewer's suggestion.

The images in Fig. 5 are the final result of overlaying the images from different layers of the 3D spheroids, thus, some images look a little over-exposed.

The following Figure 3 show the images of different layers of the 3D spheroids treated with TGFBI, which are corresponding to the middle panel in Fig. 5c. It can be found that CLDN1 is

expressed both in cell membrane and nucleus, and it seems that the high level expressions of CLDN1 in nucleus could cover its expression on cell membrane.

Figure 3.

9. Fig.6a: The proof for TGFBI knockdown should be provided, and the delayed wound healing upon TGFBI knockdown should be evidenced by the quantification data of wound closure.

Answer: Many thanks for the reviewer’s suggestion.

In the revised manuscript, we have provided the WB data of TGFBI knockdown in mouse skin tissues in Fig.S7a. In Fig.S7b and S7c, we have added the wound healing images of the mouse skin tissues and the quantitative analysis of wound healing area.

Fig. S7a-c in the revised manuscript.

10. Fig.6e: According to the morphology, the tissue sections shown in Fig.6e seem not located at epithelial tongue. Please confirm it. There is a mislabel in the lower panel: KRT14 should be PCNA.

Answer: Many thanks for the reviewer’s suggestion.

In Fig.6, we want to show the proliferation and tongue sites at day 7 and day 14. In the previous manuscript, we made a mistake in the description of figure legend. Fig. 6d and 6e represent the 14th day after wound healing, not the 7th day. Therefore, the epidermis had basically completed re-epidermization on the day 14, and the epithelial tongue could not be observed.

In the revised manuscript, we have corrected this mistake in figure legend of Fig. 6, and also revised the marker label of the lower panel of Fig. 6e to “PCNA and MMP7”.

Fig. 6 in the revised manuscript.

11. Fig.7d & Line 379: “The skin tissues in which SSP-EpSCs were present were prone to fibrosis during culture.” Cutaneous fibrosis, the result of aberrant process leading to abnormal deposition of ECM in the dermis, is commonly used to describe the underlying dermis, not epidermal cells. Did authors mean to say “EpSCs undergo epithelial-mesenchymal transition”? or did authors identify “excess production of ECM proteins by SSP-EpSCs”? Please explain it.

Answer: Many thanks for the reviewer’s suggestion.

We compared the EpSCs isolated from the skin tissues of SSP patients and normal people. By analysis of the differential expressed proteins, there were 84 ECM proteins that expressed with higher abundance in SSP patients compared with normal people, including 3 collagens, 15 glycoproteins, 1 proteoglycans, 20 ECM affiliated proteins, 34 ECM regulated proteins, and 11

secretory factors (Figure 4a). Most of these ECMs are involved in the biological processes of ECM assembly, proteolysis, epidermal cell differentiation, response to wound healing, and regulation of fibroblast migration (Figure 4b). Among them, COL18A1, LAMA5, LAMA1, and TGFβ were related to epithelial-mesenchymal transition. Therefore, from the perspective of ECM expression and cell morphology, EpSCs isolated from patient skin tissues are more difficult to maintain their stemness and lead to over differentiate.

Figure. 4

12. Fig.S4g: “Different shapes represent different signaling pathways related to the interacting proteins.”. No different shapes were presented in this interactome network. Could authors also define which proteins are involved in Wnt signaling?

Answer: Many thanks for the reviewer’s suggestion.

We mean that in Fig. S4g the proteins in different groups represents different functions. We have modified the figure legends of Fig. S4g as “Different groups represent different functions related to the interacting proteins.” In addition, the proteins involved in Wnt signaling pathway are represented using the diamond shape.

Fig. S4g in the revised manuscript.

13. Fig.S6e: Two bands are shown for TGFBI in medium fraction. Please explain it.

Answer: Many thanks for the reviewer’s suggestion.

There are some nonspecific bands in the supernatant of the previous Fig. S6e. We expanded the amount of supernatant and enriched TGFBI protein with immunoprecipitation. After cleaning the magnetic beads for many times, we reduced non-specific binding, reperformed the immunoblotting experiment on TGFBI, and replaced the previous image as Fig. S6d in the revised manuscript.

Fig. S6e in the revised manuscript.

14. Line 83-84: there are several phosphorylation sites for β -catenin; some are for degradation, e.g. S45, S33/37/T41; some are for activity and nucleus accumulation, e.g. S552. It should be defined clearly. The references should be updated to more recent articles.

Answer: Many thanks for the reviewer’s suggestion.

We have tested the phosphorylation site S33/37/T41 on β -catenin, and found it was downregulated after TGFBI treatment, and this result has been provided in Fig. 4k. Also, we have

added the descriptions of phosphorylation sites for β -catenin and updated the references in introduction.

Fig. 4k in the revised manuscript.

15. Line 290: "...syphilis infection destroyed epidermal development and...". Epidermal development usually indicates the developmental process during embryogenesis. It would be more appropriate to use "epidermal homeostasis" or "epidermal stratification".

Answer: Many thanks for the reviewer's suggestion.

We revised the statement of "epidermal development" to "epidermal homeostasis".

16. Line 343-350: No description or data interpretation on the effects of XAV939 in TGFBI-treated organoids. It remains unclear the relationship between Wnt signaling and TGFBI.

Answer: Many thanks for the reviewer's suggestion.

We added the description to interpret the effects of XAV939 in TGFBI-treated organoids, as follows: "Furthermore, the expression of maturation marker KRT10 was upregulated in 3D EpSC cultures after XAV939 treatment, even with TGFBI added (Fig. 5a), indicating that TGFBI can promote the EpSC proliferation but does not show the ability to promote EpSC differentiation."

17. Line 363-364: Aren't MMP proteins involved in cell migration rather than cell proliferation? Please confirm it.

Answer: Many thanks for the reviewer's suggestion.

MMP proteins can degrade ECM and expand the space required for cell migration. We revised the statement of "cell proliferation" to "cell migration".

18. English writing needs to be edited carefully. For examples:

- Line 313: "TGFBI expression could be induced with TGF β 1 and was secreted in the extracellular space in fibroblasts." should be "TGFBI expression could be induced by the treatment of TGF β 1 and was secreted in the extracellular space in between fibroblasts".

- Line 330: According to the diagram shown in Fig.4j, "EpSC-derived hiPSCs (EpSC-hiPSCs)" is not an accurate term; it should be "EpSCs derived from hiPSCs" or "hiPSC-derived EpSCs".

Answer: Many thanks for the reviewer's suggestion.

The sentence “TGFBI expression could be induced with TGF β 1 and was secreted in the extracellular space in fibroblasts.” has been revised to “TGFBI expression could be induced by the treatment of TGF β 1 and was secreted in the extracellular space between fibroblasts (Supplementary Fig. 6c-e).”

The description “EpSC-derived hiPSCs (EpSC-hiPSCs)” has been revised to “hiPSC-derived EpSCs”

Besides, we have checked the manuscript detailedly and corrected several language errors. Also, the manuscript has been revised by a native English speaker.

Reviewer #2 (Remarks to the Author):

I was asked to evaluate the proteomics portion of this work. The authors did an excellent job of designing the study, from the information I had available to me. All the standard tools were used for processing and analysis, and this work was done at a very high level. The authors are to be commended for doing replicate injections of replicate samples for each skin layer; that gives the absolute best/most reliable quantification in this type of experiment.

If the authors truly did not randomize their samples at all, though, that is somewhat problematic. Under these circumstances, it is difficult to conclude whether any significant changes are due to real, biological differences or due to instrument drift over time or other possible confounding effects. Also, I was not able to examine the raw data in the ProteomeXchange submission because I did not have the login information.

From the information available to me, this work appears to be very good. However, I would like to examine the data in ProteomeXchange and also have the authors address the issue of sample randomization.

Answer: Many thanks for the reviewer's approval of our work.

This study includes two groups, the secondary syphilis patients group and control donors group. Five patients and the matched controls that conformed to the inclusion criteria were randomly sampled from the biobank of the Department of Dermatology and Venereology, Peking Union Medical College Hospital. Thus, the tissue samples used in this study were randomly selected.

In "Life sciences study design" section in the "Reporting summary" document, we have filled the following information in "Randomization" column "No randomization took place during processing or analyses of tissue and cells." This means that the skin tissues from different patients or controls were processed separately; and the skin tissues from different individual didn't mixed. To avoid possible misunderstanding, we have revised the "Reporting summary" document to make the meaning more clear.

All proteomics raw data have been deposited to the ProteomeXchange Consortium via the iProX partner repository with the dataset identifier PXD027093. In the peer review process, it can be access via this link (<https://www.iprox.cn/page/PSV023.html?url=16254639404822ohA,Xp8E>); and the dataset will be public available when this paper is published.

Reviewer #3 (Remarks to the Author):

This data-rich and generally well-written manuscript describes use of tissue-specific proteome and transcriptome analysis. In general, the data are convincing. However, the results are incompletely described, and in particular the figure legends contain insufficient information for the reader to be able to interpret them. Pertinent prior studies are not incorporated in the interpretation or cited. The overall meaning of the results are also not described adequately.

The manuscript does not adequately cite or integrate prior findings. See, for example, Cruz et al., *PLoS Negl Trop Dis*. 2012;6(7):e1717. doi: 10.1371/journal.pntd.0001717. Epub 2012 Jul 17.

Answer: Many thanks for the reviewer's suggestion.

We have added the reference suggested by the reviewer. Also, we have added several other related references in the revised manuscript, including the references 2, 5, 6, 15, 16, 19, 24, and 25.

1. 1. 179-181 and elsewhere. In many places in the manuscript, varied abundance of proteins in different tissue regions is referred to as upregulation or downregulation. This terminology is incorrect, because (as exemplified by keratins in the epidermis and collagen I in the dermis) many proteins accumulate during development. Their levels do not necessarily correlate with the relative mRNA levels or ONGOING protein production in the tissue region. For example, the stratum corneum lacks nuclei and likely has little new mRNA and protein production; most of the proteins present was produced during the cells' development in the stratum spinosum and granulosum. The proteins are also subject to differential degradation and modification. Therefore it would more appropriate to refer to differences in levels of protein abundance rather than upregulation or downregulation.

Answer: Many thanks for the reviewer's suggestion.

We have taken the reviewer's suggestion and revised the description of varied abundance of proteins (lines 179-181 in the previous version), as follows: "We analyzed the proteins **with different abundance expression trends** in epidermal cells from the BL to the GS and SC. The proteins **with increased abundance** were enriched in...The proteins **with decreased abundance** from the BL...". The similar descriptions in other parts of this article have also been revised.

The descriptions of "upregulated/downregulated" kept in the text are specifically referred to the proteins or genes that are significantly differentially expressed in two compared group by statistical analysis.

2. 1. 197. Does it make sense that proteins associated with neuronal development are present in high levels in the GS layer? It would be better to examine the underlying functions of the proteins, which in this tissue would be expected to have little role in neuronal development.

Answer: Many thanks for the reviewer's suggestion.

We have modified the sense and removed the description of proteins associated with neuronal development in the text.

l. 362 and Fig. 6. The use of the term ‘epithelial tongue’ would be confusing to most readers. Alternative wording should be used.

Answer: Many thanks for the reviewer’s suggestion.

In the revised manuscript, we have revised the term “epithelial tongue” as “wound edge”.

l. 486. The equipment used for laser dissection is not described.

we added the equipment used for laser dissection.

Answer: Many thanks for the reviewer’s suggestion.

We added the equipment used for laser dissection.

The equipment used for laser dissection has been provided at the beginning of “LCM of eight layers of skin samples” section in Methods, as follows: “LCM was performed using a laser microdissection system from Molecular Machines and Industries (MMI CellCut Laser Microdissection, Eching, Germany)”.

The manuscript would benefit from a figure showing a model of how TGFBI expression is proposed to be related to epidermal development, EpSC proliferation, and expression of the proteins involved in differentiation of the layers. As it stands, the overall findings are unclear.

Answer: Many thanks for the reviewer’s suggestion.

In the revised manuscript, we have added the supplementary Fig. S8 to show the schematic diagram of TGFBI promoting skin re-epidermization during wound healing through enhancing the growth of EpSCs.

Fig. S8 in the revised manuscript.

In general, the figure legends should be sufficiently detailed so that the reader does not have to refer to the body of the manuscript to easily understand the figure. That is not the case for any of the figures. The figures would also benefit from labeling and arrows showing key features. Specific comments are provided below.

Answer: Many thanks for the reviewer’s suggestion.

In the revised manuscript, figure legends have been extended with more details.

Fig. 1. Further labeling and explanation are needed in the figure and legend. The first sentence should state “in normal human skin” or something similar. The images in 1a should have labeling for the epidermis, basement membrane and dermis, and the legend should describe the coloration of each specific label and the DAPI staining. In 1c, “Spatially” should be changed to “Spatial”, and the validation description should be expanded somewhat. In 1d legend, “Number of proteins detected” is suggested. In 1g, the meaning of “summed intensities of the proteins of interest by the summed intensities of all proteins” is unclear, in that the summed values for each region do not approach 100 (particularly for the basement membrane).

Answer: Many thanks for the reviewer’s suggestion.

We revised the legend of Fig. 1 and added more labels and explanation in figure.

The title of Fig. 1 has been revised to “Quantitative proteome profiling of spatially distinct protein signatures **in normal human skin.**”

The epidermis, basement membrane, and dermis of the images in 1a have been labeled, and the coloration of each specific label and the DAPI staining has been described in the legend.

The label “Spatially proteome” has been modified to “Spatial proteome” in Fig. 1c.

The legend of Fig. 1d has been revised to “**Number of proteins identified** in SC, GS, BL, BM, SD, and DD.”

In 1g, the percentage was calculated by dividing the summed intensities of the proteins of interest (keratins, collagens, glycoproteins, proteoglycans, ECM regulators, ECM-affiliated proteins, or secreted proteins) by the summed intensities of all proteins identified in specific skin layer. Except for the seven kinds of proteins counted in Fig. 1g, there are many other proteins identified in each skin layer. Thus, the summed percentages of these seven kinds of proteins do not approach 100.

Taking the stratum corneum (SC) as an example, a total of 4686 proteins identified in SC with the sum protein intensities of $1.65\text{E}+09$. Within the 4686 proteins, 50 keratins, 28 collagens, 82 glycoproteins, 15 proteoglycans, 93 ECM regulators, 39 ECM-affiliated proteins, and 40 secreted proteins were identified. The summed intensities of the keratins were $4.51\text{E}+08$, then, the percentage for keratins in SC was calculated as: $4.51\text{E}+08/1.65\text{E}+09*100\%=27.31\%$.

Fig. 2. For the 2a legend, it should be indicated that each column within the skin regions corresponds to a different human skin specimen (if that is the case). Why is the number variable? Perhaps dashed lines could be drawn at the upper and lower boundaries to more clearly associate the regions of the heat map with the descriptions on the right side. In 2c, the EGFR and COL4AJ specimens appear to be IHC rather than immunofluorescence, and the color difference between the specific staining and nonspecific stain (hematoxylin?) is not sufficient to discern. These should be redone. As for 1a, 2c should be more thoroughly described in the legend; perhaps arrows could be added to indicate areas of specific staining.

Answer: Many thanks for the reviewer’s suggestion.

In Fig. 2a legend, we have indicated the meaning of each column as follows: “Each column within the skin regions corresponds to a different human skin specimen.”

The skin tissues from five healthy donors were used as control groups. We performed the laser capture microdissection (LCM) on the frozen embedded sections to get different skin layers with exact localization. As not all five human skin specimen had enough frozen sections for all skin layers, thus, the number variable in the heatmap of Fig. 2a. However, at least three samples were included in each skin regions ($n \geq 3$).

The modules 1-6 have been labeled in the heatmap to associate the regions with the descriptions on the right side.

In Fig. 2c, we have added the dashed lines, redone the staining experiments of EGFR and COL4A1, and added the arrows to indicate areas of specific staining.

Fig. 3. In 3a, what is the staining shown under “Treponema pallidum”? The staining method is not described. Also, the coloration in the lower right panel of 3a does not match that of the low magnification view. Scale bars should be used consistently and labeled. In 3b legend, SSP-GS should be described as the stratum granulosum-spinosum layers rather than stratum corneum. For 3d legend, the term “gradually” does not fit here. In 3e, the NC and SSP groups should be more clearly separated in the figure.

Answer: Many thanks for the reviewer’s suggestion.

Fig. 3a shows the immunohistochemistry staining of *Treponema pallidum* infected skin. The detailed staining method has been provided in Methods as follows: “4- μ m-thick sections of each biopsy specimen were cut from formalin-fixed and paraffin-embedded tissue blocks. Following heat-induced epitope retrieval and primary rabbit polyclonal antibody directed to *Treponema pallidum* (Novus Biologicals, Inc., CA, USA), Immunohistochemistry (IHC) staining was performed on BenchMark ULTRA automated staining instrument (Roche Diagnostics) using ultraView Universal Alkaline Phosphatase Red Detection Kit (Roche)”

We have corrected the coloration labels in Fig.3a. In order to present the expression levels of KRT10 and KRT14 better, the DAPI (blue) wasn’t shown in low magnification view.

The scale bars have been labeled with the images in the revised manuscript.

We have corrected the description of SSP-GS as “granular-spinous” in Fig. 3b legend.

The NC and SSP groups have been clearly separated in Fig. 3e by a black line of dashes.

Fig. 3a in the revised manuscript.

Fig. 4. The title sentence should indicate that the data in this figure is based primarily on RNA-seq analysis, as opposed to protein analysis in the preceding figures. In 4a, the coloration and the meaning of the dashed line should be described. In 4b legend, it should be stated that these are in vitro cultures of EpSCs. The treatment interval (hours) should be indicated. For 4h, the x axis should be labeled, and a more standard y axis (rather than e5) should be used. Fig. 4j is unclear and should be omitted.

Answer: Many thanks for the reviewer’s suggestion.

We have modified the title of Fig. 4 to indicate that the data in this figure is based primarily on RNA-seq analysis, as follows: “TGFBI enhanced proliferation of EpSCs through the wnt/ β -catenin pathway based on the transcriptome analysis.”

The description for the coloration and the meaning of the dashed line in Fig. 4a has been added as follows: “The white dashed lines represent the BM zone of the skin tissues.”

In Fig. 4b legend, the in vitro cultures of EpSCs has been stated and the treatment interval (hours) has been indicated as follows: “Biological process analysis of the upregulated genes in the TGFBI treated (100 ng/ml) EpSCs of primary culture for 48 hours, compared to the control group.”

Besides, we have replaced Fig. 4h with Fig. 4j in the revised manuscript, and removed the previous Fig. 4j.

Fig. 4j in the revised manuscript.

l. 248 and elsewhere. expression rather than expressions.

Answer: Many thanks for the reviewer’s suggestion.

We revised “expressions” to “expression”.

REVIEWER COMMENTS

Reviewer #1 (Remarks to the Author):

Authors addressed all of my concerns; however, a few minor points are required to be revised before the manuscript can be accepted.

1. Quantification data for immunostaining analyses shown in Fig. 4g, 4k, 4o, 5a, 5c, 5d and 7c. What are those numbers at the corner of Fig. 4g, 4k, 4o, 5a, 5c, 5d, 7b and 7c referred to? Numbers or percentage of positive cells per image or per spheroid? Given that cell numbers could vary a lot from one image to another, it is more accurate to use percentage to present the quantification data. Please define it in the figure legends accordingly.
2. Revised images in Fig. 3f should indicate basement membrane by dotted lines as those shown in Fig. 4a. Even though the authors had changed a new image, the staining for TGFBI in Fig. 3f remains highly positive to the epidermis. It seems overlapping with KRT14 staining. Please double confirm it by performing new staining.
3. Please explain how the authors quantified E-cad⁺ cells shown in Fig. 4h? Almost 100% of cells with E-cadherin staining at cell-cell junction. It is surprising to see only 20-30 presented in the images unless the numbers are referred to the cell number. If this is the case, it should be re-calculated and presented as percentage.
4. As for Fig. 5, it will be helpful if the Z-stack images that are shown in Figure 2 (rebuttal letter) can be provided in the supplementary figures.
5. As for western blotting analysis, both phosphorylated and total proteins should be provided for assessment. p-GSK3b-S9 needs to be provided in Fig. 4k. Total GSK3b protein is missing in Fig. S6i. p-b-catenin S33/37/41 is not provided in Fig. S6j. The loading control for nuclear protein, e.g. laminin, is required for Fig. 4l.

Reviewer #2 (Remarks to the Author):

I thank the authors for responding to my question regarding randomization, although I wasn't clear enough about what I was asking. Were the mass spec samples run at the same time, and was a randomized run order used? Were the samples all run at different times? The response sounds as though samples from each individual patient, regardless of condition, were all processed and analyzed at completely different times. This could impact results for quantification, especially if all controls were run and then all with secondary syphilis (for example). If samples from all patients were run at different points in time, were the different layers analyzed in the same order every time or were they randomized? All of these factors could impact the results. Normalization helps to some extent, but I'm not sure if it would completely compensate for samples being analyzed at completely different points in time and in the exact same order every time.

Reviewer #3 (Remarks to the Author):

The manuscript represents a detailed, major work on the distribution of proteins in the skin using a combination of mass spectroscopy, immunolabeling, and other approaches. It has been revised extensively from its prior version, and it appears that the most of the reviewers' comments have been addressed effectively. Very minor grammatical issues remain, including the improper use of the word

"contract" (construct?) on l. 150 and the common use of the word "expressions" where "expression" is appropriate.

REVIEWER COMMENTS

Reviewer #1 (Remarks to the Author):

Authors addressed all of my concerns; however, a few minor points are required to be revised before the manuscript can be accepted.

1. Quantification data for immunostaining analyses shown in Fig.4g, 4k, 4o, 5a, 5c, 5d and 7c.

What are those numbers at the corner of Fig. 4g, 4k, 4o, 5a, 5c, 5d, 7b and 7c referred to?

Numbers or percentage of positive cells per image or per spheroid? Given that cell numbers could vary a lot from one image to another, it is more accurate to use percentage to present the quantification data. Please define it in the figure legends accordingly.

Answer: Many thanks for the editor's suggestion.

The protein expression level of immunofluorescence staining experiment can be measured by the intensity of fluorescence. Thus, we used the Image J software to detect the fluorescence intensities and quantify the protein expressions. For each single channel (monochrome) fluorescence picture, the gray value of each pixel represents the fluorescence intensity of the point. The fluorescence intensity of a specific region is calculated as follows: Average fluorescence intensity (Mean) % = Sum of fluorescence intensity of the region (IntDen) / Area of the region (Area)*100%.

Those numbers at the corner of Fig. 4f, 4h, 5a, 5c, 5d, 7b and 7c referred to the percentage of the average fluorescence intensity of the specific protein (fluorescent channel); we are sorry for omitting the percent symbol (%) in the previous figures. In this revised version, we have corrected this mistake and added the percent symbol (%) for quantified data in Fig 4f, 4h, 5a, 5c, 5d, 7b and 7c.

Fig. 4f, h in the revised manuscript.

Fig. 5 in the revised manuscript.

Fig. 7b, c in the revised manuscript.

2. Revised images in Fig. 3f should indicate basement membrane by dotted lines as those shown in Fig. 4a. Even though the authors had changed a new image, the staining for TGFB1 in Fig. 3f remains highly positive to the epidermis. It seems overlapping with KRT14 staining. Please double confirm it by performing new staining.

Answer: Many thanks for the editor's suggestion.

We performed new staining of TGFB1 and KRT14 in health and SSP skins and updated the staining results in Fig. 3f. TGFB1 is mainly expressed around the basal stem cell layer (BL layer) and at the bottom of the BL layer, indicating that TGFB1 is mainly located around the basement membrane after secreted by cells. KRT14 is also mainly expressed in the basal stem cell layer, and secreted outside the cells. Therefore, TGFB1 and KRT14 are almost co-expression in the skin. In addition, the basement membrane has been indicated with dotted lines in Fig. 3f.

Fig. 3f in the revised manuscript.

3. Please explain how the authors quantified E-cadherin+ cells shown in Fig. 4h? Almost 100% of cells with E-cadherin staining at cell-cell junction. It is surprising to see only 20-30 presented in the images unless the numbers are referred to the cell number. If this is the case, it should be re-calculated and presented as percentage.

Answer: Many thanks for the editor's suggestion.

The fluorescence intensity was used to quantify the protein expression level by Image J analysis software. For ECAD single channel pictures, the threshold was adjusted first; then the appropriate threshold algorithm (Default) was selected and applied to all pictures in different groups. Next, the parameters were set to ensure that the measured value is just for the proteins expressed in the specific region. Finally, the percentage of average fluorescence intensity of ECAD (%) is calculated by the following formula: Average fluorescence intensity (Mean) % =

Sum of fluorescence intensity of the region (IntDen) / Area of the region (Area) *100%..

The numbers in Fig. 4f represented the percentage of the average fluorescence intensity of the specific protein (fluorescent channel); we forgot to add the percent symbol (%) in the previous figures. In this revised version, we corrected this mistake and added the percent symbol (%) for quantified data in Fig 4f.

Fig. 4f in the revised manuscript.

4. As for Fig. 5, it will be helpful if the Z-stack images that are shown in Figure 2 (rebuttal letter) can be provided in the supplementary figures.

Answer: Many thanks for the editor's suggestion.

In this revised manuscript, several representative Z-stack images of 3D spheroids have been provided as Supplementary Fig. 7a to support the results in Fig. 5.

Supplementary Fig. 7a in the revised manuscript.

5. As for western blotting analysis, both phosphorylated and total proteins should be provided for assessment. p-GSK3 β -S9 needs to be provided in Fig. 4k. Total GSK3 β protein is missing in Fig. S6i. p-b-catenin S33/37/41 is not provided in Fig. S6j. The loading control for nuclear protein, e.g. laminin, is required for Fig. 4l.

Answer: Many thanks for the editor's suggestion.

We have added the western blotting results of p-GSK3 β -S9 in Fig. 4k, total GSK3 β protein in Supplementary Fig. 7c (Supplementary Fig. 6i in the previous version), and p- β -catenin S33/37/41 in Supplementary Fig. 7d (Supplementary Fig. 6j in the previous version), as well as the loading control (laminin) for nuclear protein in Fig. 4l.

Fig. 4k, l in the revised manuscript.

Supplementary Fig. 7c, d in the revised manuscript.

Reviewer #2 (Remarks to the Author):

I thank the authors for responding to my question regarding randomization, although I wasn't clear enough about what I was asking. Were the mass spec samples run at the same time, and was a randomized run order used? Were the samples all run at different times? The response sounds as though samples from each individual patient, regardless of condition, were all processed and analyzed at completely different times. This could impact results for quantification, especially if all controls were run and then all with secondary syphilis (for example). If samples from all patients were run at different points in time, were the different layers analyzed in the same order every time or were they randomized? All of these factors could impact the results. Normalization helps to some extent, but I'm not sure if it would completely compensate for samples being analyzed at completely different points in time and in the exact same order every time.

Answer: Many thanks for the editor's suggestion.

All skin tissue samples prepared for mass spectrometer (MS) were processed simultaneously. To avoid the error caused by different MS instrument, all peptides samples were run use the same HF-X Orbitrap MS in a sequential order at different time. The tissue samples preparation and peptide mixtures analysis by MS were processed by two different investigators. Both investigators were kept blinded to the group (control or secondary syphilis) and layer (skin layers) information. Thus, all samples were analyzed using exactly the same procedures and in randomized orders on MS instrument.

For the quality control of the performance of MS platform, the HEK293T cell lysate was measured every two days as the quality-control standard sample. The HEK293T standard sample was digested and analyzed using the same method, condition, and MS instrument as the skin samples. In the whole MS experiment procedures, six HEK293T standard samples (Sample 1-6) were analyzed and generated six MS datasets. As shown in Figure 1, the six replicated standard samples identified similar protein numbers (Figure 1a), and more than 90% of the proteins (4722, Figure 1b) were identified by all samples. A pairwise spearman correlation coefficient was calculated for protein quantification of all quality-control samples, the average correlation coefficient among the standards was 0.98. All these results demonstrated the consistent stability of the mass spectrometry platform. Also, we have added the detailed information about this quality control process in "MS analysis" section in Methods.

Figure 1. Quality control of mass spectrometry platform.

Reviewer #3 (Remarks to the Author):

The manuscript represents a detailed, major work on the distribution of proteins in the skin using a combination of mass spectroscopy, immunolabeling, and other approaches. It has been revised extensively from its prior version, and it appears that the most of the reviewers' comments have been addressed effectively. Very minor grammatical issues remain, including the improper use of the word "contract" (construct?) on l. 150 and the common use of the word "expressions" where "expression" is appropriate.

Answer: Many thanks for the editor's suggestion.

We have corrected the word "contract" to "construct".

We have taken the reviewer's suggestion and use the common use of the word "expression".

REVIEWERS' COMMENTS

Reviewer #1 (Remarks to the Author):

Authors revised the manuscript accordingly; however, they should pay more attention on the detail of their revision. Below are the minor points that should be improved.

1. The method for immunostaining quantification should be described in the section of Methods.
2. Fig. 3f: If TGFBI is expressed by the basal cells and secreted to the basement membrane, the images in the previous manuscript would be more convincing. Please note that KRT14 is an intermediate filament protein, a structure protein of a cell that is impossible to be secreted outside of the cell.
3. Fig. 4l: GAPDH should be used to detected the cytoplasmic fraction, not the total protein extract.
4. Fig. S7c, d: Total GSK3 β remains missing in S7c, and total b-catenin should be shown after the p-b-catenin in S7d. Authors should pay more attention to their revision.

Reviewer #2 (Remarks to the Author):

The authors have addressed my concerns.

REVIEWER COMMENTS

Reviewer #1 (Remarks to the Author):

Authors revised the manuscript accordingly; however, they should pay more attention on the detail of their revision. Below are the minor points that should be improved.

1. The method for immunostaining quantification should be described in the section of Methods.

Answer: Many thanks for the editor's suggestion.

The method for immunofluorescence quantification has been described in the “Confocal microscopy” section of Methods, as follows: “The protein expression level of immunofluorescence staining experiment can be measured by the intensity of fluorescence. The ImageJ software (version 1.53) was used to detect the fluorescence intensities and quantify the protein expressions. For each single channel (monochrome) fluorescence picture, the gray value of each pixel represents the fluorescence intensity of the point. The fluorescence intensity of a specific region is calculated as follows: Average fluorescence intensity (Mean) % = Sum of fluorescence intensity of the region (IntDen) / Area of the region (Area)*100%.”

2. Fig. 3f: If TGFBI is expressed by the basal cells and secreted to the basement membrane, the images in the previous manuscript would be more convincing. Please note that KRT14 is an intermediate filament protein, a structure protein of a cell that is impossible to be secreted outside of the cell.

Answer: Many thanks for the editor's suggestion.

According to our previous experiments, TGFBI is expressed in dermal fibroblasts after TGF β stimulation, while TGFBI is almost not expressed in epidermal cells (Figure 1). We believe that TGFBI protein is secreted by fibroblasts first and then is located around the basement membrane to promote the function of epidermal stem cells. We're sorry for misusing the word “expressed” in the last response letter. It should be “located” instead of “expressed”.

Figure 1.

It is known that keratin intermediate filaments, the major components of epithelial cells, are encoded by a large family of 54 conserved genes which can be regulated individually in a tissue- and differentiation-specific fashion. Among them, type I keratin 14 (KRT14) and type II keratin 5 (KRT5) can co-polymerize and form the prominent intermediate filaments apparatus which occurs in the progenitor basal layer of epidermis and related complex epithelia. The above process can

provide structural support and mechanical resilience to keratinocytes in the basal layer of epidermis and related epithelia. In fact, our group has been concerned about the expression and localization of keratins for a long time. Based on the biological annotation from public protein database and our previously experiment results, it is suggested that KRT14 can be secreted outside of the cell. First, according to the cellular component annotation of UniProt database (<https://www.uniprot.org/uniprot/P02533>), the subcellular locations of KRT14 include extracellular region or secreted and extracellular exosome. Second, our previous study confirmed that part of KRT14 protein was expressed outside the epidermal stem cells, and may be transmitted outside the cells by exosomes (Leng et al., 2020), which is consistent with other study (Gonzalez-Begne et al., 2009). As shown in Figure 2, the expression of KRT14 is not in a horizontal plane during shooting, which also happens to KRT10 in many experiments. Also, the western blot analysis demonstrated that KRT14 can expressed both in EpSCs and EpSC-derived vesicles (Figure 3), indicating that KRT14 could be secreted outside of the epidermal stem cells by exosome. So far, the role of these secreted keratins is not clear; but in some studies, these extracted keratins were used as scaffolds to culture cells and it was found that they can promote cell growth and wound healing (Reichl, 2009).

[redacted]

3. Fig. 4i: GAPDH should be used to detected the cytoplasmic fraction, not the total protein extract.

Answer: Many thanks for the editor's suggestion.

We have updated Fig. 4i and GAPDH has been used to detect the cytoplasmic fraction in this experiment.

Fig. 4i in the revised manuscript.

4. Fig. S7c, d: Total GSK3β remains missing in S7c, and total b-catenin should be shown after the p-b-catenin in S7d. Authors should pay more attention to their revision.

Answer: Many thanks for the editor's suggestion.

The experiment results of PCNA, total GSK3β, and phosphorylated GSK3β in Supplementary Fig. S7c have been presented in the Supplementary Fig. 7d and Fig. 4k, thus, we deleted the original Supplementary Fig. S7c in this revised manuscript.

The original Supplementary Fig. 7d has changed to Supplementary Fig. 7c, and the western blot result of total β -catenin was presented after the p- β -catenin.

Supplementary Fig. 7c in the revised manuscript.

Reference:

Gonzalez-Begne, M., Lu, B., Han, X., Hagen, F.K., Hand, A.R., Melvin, J.E., and Yates, J.R. (2009). Proteomic analysis of human parotid gland exosomes by multidimensional protein identification technology (MudPIT). *J Proteome Res* 8, 1304-1314.

Leng, L., Ma, J., Lv, L., Wang, W., Gao, D., Zhu, Y., and Wu, Z. (2020). Both Wnt signaling and epidermal stem cell-derived extracellular vesicles are involved in epidermal cell growth. *Stem Cell Res Ther* 11, 415.

Reichl, S. (2009). Films based on human hair keratin as substrates for cell culture and tissue engineering. *Biomaterials* 30, 6854-6866.